

# Uncertainty Assessment of Satellite Remote Sensing-based Evapotranspiration Estimates: A Systematic Review of Methods and Gaps

Bich Tran[1,2], Johannes van der Kwast[1], Solomon Seyoum[1], Remko Uijlenhoet[2], Graham Jewitt[2,3], Marloes Mul[1]

[1]Land and Water Management Department, IHE Delft Institute for Water Education, Delft, 2611 AX, the Netherlands
[2]Department of Water Management, Delft University of Technology, Delft, 2628 CN, the Netherlands
[3]Water Resources and Ecosystems Department, IHE Delft Institute for Water Education, Delft, 2611 AX, the Netherlands

*Correspondence to*: Bich N. Tran (b.tran@un-ihe.org)

**Abstract.** Satellite remote sensing (RS) data are increasingly being used to estimate total evaporation or evapotranspiration (ET) over large regions. Since RS-based ET (RS-ET) estimation inherits uncertainties from several sources, many available studies have assessed these uncertainties using different methods and reference data. However, the suitability of methods and reference data subsequently affects the validity of these evaluations. This study summarizes the status of the various methods applied for uncertainty assessment of RS-ET estimates, discusses the advances and caveats of these methods, identifies assessment gaps, and provides recommendations for future studies. We systematically reviewed 601 research papers published from 2011 to 2021 that assessed the uncertainty or accuracy of RS-ET estimates. We categorized and classified them based on (i) the methods used to assess uncertainties, (ii) the context where uncertainties were evaluated, and (iii) the metrics used to report uncertainties. Our quantitative synthesis shows that the uncertainty assessments of RS-ET estimates are not consistent and comparable in terms of methodology, reference data, geographical distribution, and uncertainty presentation. Most studies used validation methods using Eddy Covariance (EC) based ET estimates as reference. However, in many regions such as Africa and the Middle East, other references are often used due to the lack of EC stations. The accuracy and uncertainty of RS-ET estimates are most often described by Root-Mean-Squared Error (RMSE). When validating against EC-based estimates, the RMSE of daily RS-ET varies greatly among different locations and levels of temporal support, ranging from 0.01 to 6.65 mm/day with a mean of 1.12 mm/day. We conclude that future studies need to report the context of validation, the uncertainty of the reference datasets, the mismatch in temporal and spatial scales of reference datasets to that of the RS-ET estimates, and multiple performance metrics with their variation in different conditions and statistical significance to provide a comprehensive interpretation to assist potential users. We provide specific recommendations in this regard. Furthermore, extending the application of RS-ET to regions that lack validation will require obtaining additional ground-based data and combining different methods for uncertainty assessment.



## 1 Introduction

Evapotranspiration (ET) is the key variable linking the water, energy, and carbon cycles of the Earth. In the continental water cycle, it is the second-largest flux after precipitation (Korzoun et al., 1978), which predominates the demand side of water resources. It is associated with latent heat flux in the surface energy balance. ET combines evaporation of water from soil, free water surfaces and plants and thus, depends on many factors, such as the atmospheric and vegetation conditions, the availability

of water in the soil, water bodies, canopy, and surface roughness. The complexity of measuring ET directly makes it difficult and expensive to routinely measure and capture its spatial variation, as this requires a dense network of *in-situ* gauging stations. Therefore, satellite remote sensing (RS) observations have been increasingly used for estimating ET spatially.

As ET cannot be directly measured by sensors from space, retrieval algorithms or models are needed to estimate ET from other variables observed by RS. These models estimate ET from optical and/or thermal RS data, and include now well-known models

such as SEBAL (Bastiaanssen et al., 1998), TSEB (Kustas and Norman, 1999), SEBS (Su, 2002), METRIC (Allen et al., 2007), and ALEXI (Anderson et al., 2011). These diverse models, parameters, input RS data sources, and processing techniques result in a wide range of RS-based ET estimates (Jimenez et al., 2011, Long et al., 2014, Chen et al., 2014).

While many studies have evaluated the performance of RS-based ET (RS-ET) models, none of them has concluded that a single model performs best in all situations (e.g., Ferguson et al., 2010; Vinukollu et al., 2011a). Furthermore, the accessibility

and uncertainty lead to significant challenges to operational applications (e.g., irrigation scheduling and drought monitoring) using output from these models. For some models, retrieving ET estimates requires access to, and expertise in, these models. Driven by community needs, several projects have provided platforms to increase public access to various RS-ET data products. These projects and outputs include MODIS16 (Mu et al., 2011), SSEBop (Senay et al., 2013), GLEAM (Miralles et al., 2011a), WaPOR (FAO, 2018) and OpenET (Melton et al., 2021).

With the evolution of these now widely accessible RS-ET data products, characterizing and quantifying the uncertainties associated with these data products becomes important for data users (i.e., water managers and policymakers). Uncertainty assessment helps data users know what level of confidence they can have in ET estimates and the inferred information about water resources (e.g., crop water consumption, water depletion).

Previous reviews have discussed RS-ET estimates and uncertainty, which are relevant to this review (Figure 1). Table S2 in

Supplementary Information summarizes the foci of these reviews. Many of these reviews focused on outlining the methods to estimate ET using RS-based models (e.g., Kustas and Norman, 1996; Courault et al., 2005; Wang and Dickinson, 2012; Zhang et al., 2016) and sometimes discussed the uncertainties in the estimation (Kalma et al., 2008; Glenn et al., 2011; Karimi and Bastiaanssen, 2015). However, none of these explored how uncertainties of RS-ET estimates are currently being assessed, which is an important issue in remote sensing and spatial data production (Bielecka and Burek, 2019; Wu et al., 2019; Mayr

et al., 2019). In an overview of global RS-based Essential Climate Variables (ECVs), Bayat et al. (2021) concluded that RS-ET data products lack a good practice protocol for operational validation, compared to other ECVs. Meanwhile, *in-situ*



measurements of ET also suffer from errors and uncertainty (Allen et al., 2011a) and, thus, require complete documentation to ascertain the expected accuracy and representativeness of the reported ET estimates (Allen et al., 2011b).

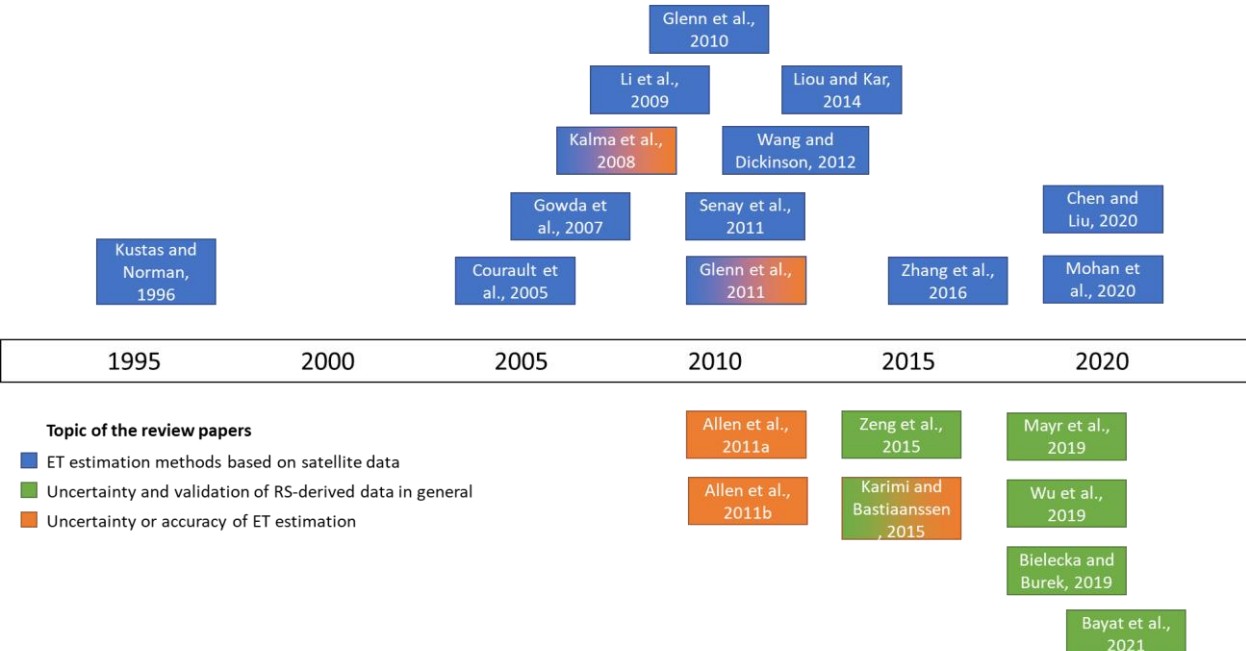


**Figure 1: Previous literature reviews on RS-ET estimation, uncertainty, and validation of RS-derived data.**

These reviews highlight the need to better advance the uncertainty assessment of RS-ET, leading to the following research questions:

- What are the common and emerging methods used to assess uncertainty in RS-ET estimates?
- In which contexts are the uncertainties of RS-ET assessed with these methods?
- What is the typical range of uncertainty in RS-ET estimates globally based on previous studies?

To answer these questions and build on existing literature, we surveyed previous studies that assessed the uncertainty or accuracy of RS-ET models or the output data products of these models. Given that many literature reviews on the uncertainty or accuracy of ET estimation have been published until 2011 (Figure 1), we focus on the period from 2011 to evaluate whether
the studies in this period adopted the valuable contributions and recommendations from these previous reviews. Given the growing volume of literature published in the field, we followed a systematic quantitative review approach to avoid subjectivity or a bias towards particular products, authors or approaches. We identified research articles with a set of predetermined criteria and categorized these articles based on (i) the methods used to assess uncertainties, (ii) the context where uncertainties were evaluated, and (iii) the metrics used to report uncertainties. We then quantified the number of articles per category to identify
any trends or gaps in literature. Furthermore, we appraised the advances and caveats of the existing methods and provided recommendations for future studies.



The rest of this paper is organized as follows: Section 2 provides the theoretical basis for the research and clarifies the key terms that we use to analyze literature using the methods described in Section 3. The results of the literature analysis, concerning assessment methods and the context when these methods are used are discussed in Section 4 and 5. Based on the
categorized literature, Section 6 discusses the use of uncertainty metrics and shows the typical range of uncertainty in RS-ET estimates. Finally, Section 7 summarizes the key points and recommendations for future research.

## 2 Theoretical frameworks

### 2.1 Uncertainty definition and representation

Uncertainty is generally defined as the state of being not completely confident or sure of something. The terms 'error',
'accuracy', 'bias', and 'precision' are sometimes used to characterize the uncertainty associated with a measurement or model output. In this study, we consider all these terms as quantifiable information about what is certain or uncertain. However, it is important to clarify these terms since they are different from 'uncertainty' (Foody and Atkinson, 2003; Heuvelink, 1998; Loew et al., 2017).

Error represents the difference between what is measured and its true value in metrology (i.e., the science of measurement)
(JCGM, 2012). The true value is the exact (but mostly unknown) value according to the theoretical definition of the variable being measured or estimated. If a true value is perfectly known, the error of a measurement is surely known. Then, there cannot be any uncertainty because we are certain of all values. Therefore, uncertainty is associated with a set of unknown true values and unknown errors. Since the true value of a quantity intended to be measured or estimated is practically unknown, 'error' and 'uncertainty' are often used interchangeably (e.g. in the work of Heuvelink, 1998).

The definition of 'accuracy' seems more disputable than 'error'. Loew et al. (2017) used a definition grounded in metrology (JCGM, 2012) and noted that accuracy is not given a numerical value. Meanwhile, Foody and Atkinson (2003) defined accuracy as "the expectation (i.e., expected value) of overall error". Both Foody and Atkinson (2003) and Heuvelink (1998) considered the Root Mean Squared Error (RMSE) as a measure of accuracy. 'Bias' (i.e., the difference between the expected value of an estimator and the true value of a parameter) is considered a measure of inaccuracy. High bias indicates inaccuracy
and low bias indicates better accuracy. Likewise, we consider standard deviation and variance as numerical quantities of imprecision since these both describe the spread of errors around the mean error.

Because uncertainty is related to unknown errors in many fields of science, it is often described using probability distributions (Montanari, 2007; Foody and Atkinson, 2003). In metrology, measurement uncertainty is a non-negative parameter that determines the probability distribution of the (possible) values attributed to the measurand (i.e., the quantity intended to be
measured) (JCGM, 2012; Loew et al., 2017). Uncertainty can also be characterized by the probability distribution of the error (instead of the measured value) (Povey and Grainger, 2015). The uncertainty (described by inaccuracy and imprecision) of a measuring instrument can be estimated from repeated measurements of a known quantity (standard reference), before





measuring an unknown quantity. Figure 2 illustrates how other terms are related to uncertainty (i.e. the probability distribution of measured value or error).

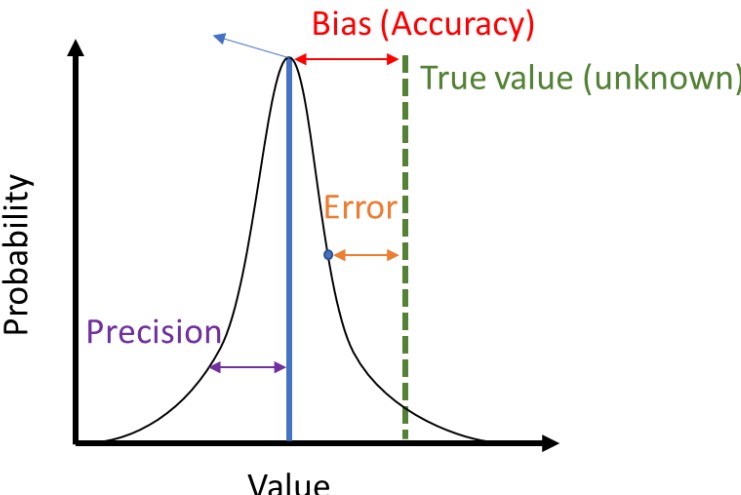

**Figure 2: Uncertainty as described by the probability distribution of measured values. Adapted from Povey and Grainger (2015) and JCGM (2012).**

Uncertainty cannot always be described using a probability distribution function in modeling or measurement. Povey and Grainger (2015) called these types of uncertainty in remote sensing the 'known, unquantifiable unknowns' (i.e., what we know to exist but are not able to quantify) and the 'unknown unknowns' (i.e., what we do not know to exist because we cannot observe). The suitability of probability theory to quantify uncertainty has been intensively discussed by many authors in hydrological science (e.g., Beven, 2016; Nearing et al., 2016). For example, Nearing et al. (2016) argued that there is epistemological uncertainty preceding the moment at which we select probability theory as the framework (i.e., our choice of epistemology) to estimate epistemic uncertainty (i.e., what we do not know certainly). Their classification of uncertainty includes philosophical and linguistic types of uncertainty, which are hardly quantifiable. Quantifiable errors are a consequence of intrinsic variability in the observation and propagated errors from auxiliary data. The 'unknown' errors are due to model approximation, simplification of reality, and sampling resolution of an instrument. Typically, uncertainty reported with satellite remote sensing data only represents the quantifiable errors but not the 'unknown' errors (Povey and Grainger, 2015).

## 2.2 Sources of uncertainties in RS-ET data production

The ET are not measured by sensors but result from models or reanalyzes, and thus, RS-ET data products are considered high level of processing (ESA, 2021; NASA, 2021). Satellite sensors measure the radiation reflected or emitted from the earth's surface or atmosphere in different regions of the electromagnetic spectrum. It should also be noted that 'directly measured'



also involves a measurement model (i.e., the way we relate the numerical output of a measuring instrument to the underlying physical state of the object that we want to measure). Raw digital imagery acquired by sensors on satellites must undergo a

chain of processing and analysis to derive useful data and information for applications (Figure 3). Therefore, the uncertainty of RS-ET data products is strongly linked with the total uncertainty in the model and in the used satellite data.

Uncertainty in input data and model parameters can propagate through model to output. In addition, uncertainty due to the change of spatial and temporal support (i.e., the area and time period over which a variable is measured), and gap filling can occur during pre-processing or post-processing steps depending on the models. Modeled estimates are typically validated with

a reference that is considered the "true" value. The differences between the former and the latter is referred to as 'compound uncertainty' in this research since it compounds all sources of uncertainty. Meanwhile, if the modeled estimates are compared with other equivalent estimates, the differences are considered 'relative uncertainty'. Since reference data is imperfect, compound uncertainty is essentially relative uncertainty except that we trust our reference to represent "true" value more than other estimates.

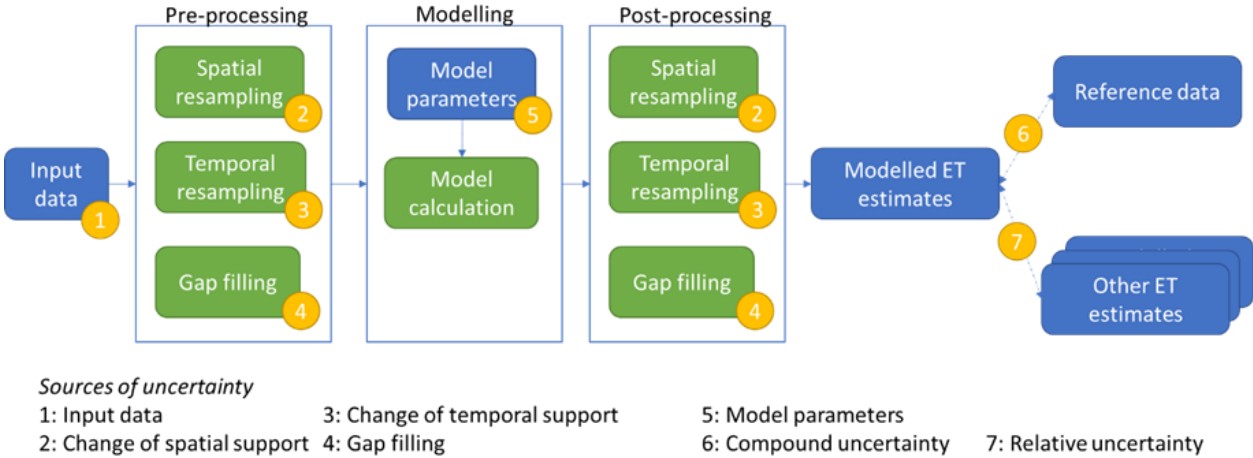


**Figure 3: The sources of uncertainty in E$_T$ estimates from the typical workflow in remote sensing-based models.**

### 2.3 Uncertainty assessment

Uncertainty assessment refers to the estimation of quantifiable uncertainties. The uncertainty from input factors (e.g., parameters and input data) can be quantified with uncertainty analysis, also called uncertainty propagation or error propagation

(Crosetto et al., 2001; Heuvelink, 1998; Wadoux et al., 2020). For RS-based models, analytical techniques which are based on the law of variance propagation (Taylor, 1997) are often not suitable because these models are essentially nonlinear functions. Numerical techniques that use such as the Monte Carlo method (MCM), numerical approximation for uncertainty propagation are generally applicable to RS-based models (Heuvelink, 1998, Crosetto et al., 2001). In MCM, the model inputs are randomly sampled from their distributions and fed into the model to generate outputs repeatedly. The variance of the output distribution

will be then considered the uncertainty in the model output (i.e., ET estimate) associated with the input variables.



The contribution of each input factor to the total uncertainty in the model output is determined by sensitivity analysis (Crosetto et al., 2001; Saltelli et al., 2021). Such analysis is primarily used to identify the factors that contribute most to the model uncertainty (Saltelli et al., 2019). There are two main approaches to sensitivity analysis (SA): local and global sensitivity analysis. Local SA (LSA) defines the model's sensitivity to an input factor (e.g., parameter or variable) as the first-order partial

derivative of the model with respect to this input factor. In contrast, global SA (GSA) explores the whole variation range of the input factors. Several global SA techniques have been developed and used in scientific modeling (Razavi and Gupta, 2015; Saltelli et al., 2008).

Due to large volumes of data and spatio-temporal dimensions, validation is often applied to confirm a data product's fit-for-purpose instead of sensitivity analysis and uncertainty analysis of its retrieval model (Crosetto et al., 2001). Validation, by

common sense, is proving the validity of something. However, the definition of validation in modeling has changed over time and is context-dependent (Bellocchi et al., 2011). The RS-derived data products are often the output of models, but validation often only refers to the data itself and not the model (Bayat et al., 2021; Loew et al., 2017; Wu et al., 2019a). Validation of RS-derived data products involves quantifying the accuracy compared to a reference (often *in-situ* datasets), which proves the validity of the data for its intended application.

Since RS-ET retrieval models can be used with different sets of satellite data, model validation and data (i.e., model result or output) validation should be distinguished. Scientific models are often considered hypotheses, and thus, can only be falsified rather than proven. Therefore, model validation does not prove the model is true but rather proves that it is empirically adequate by evaluating its relative performance with respect to observations, other models, and theoretical expectations (Oreskes et al., 1994). A valid model means that the model does not contain known or detectable flaws and is internally consistent. Meanwhile,

validation of model results depends on the quality and quantity of input parameters and the accuracy of auxiliary hypotheses that were used to derive them (Oreskes et al., 1994).  If a model was validated using a set of input satellite data, it does not guarantee that the model output will have the same quality when it is applied using a different dataset. Therefore, RS-derived data product validation is still needed even if the retrieval model has been "validated" before.

## 3 Systematic quantitative literature review method

In this literature review, we specifically focus on how the quantifiable uncertainty in the RS-ET estimate has been assessed in recent years (2011-2021). There are various literature review methods that differ in search, appraisal, synthesis, and analysis approach (Grant and Booth, 2009). The systematic quantitative literature review (SQLR) method described by Pickering and Byrne (2014) can consider a large body of research literature tby applying a systematic search and literature categorization and quantification. The quantitative results from SQLR provide subjective evidence of the trends and gaps in the literature.

Therefore, this approach was selected to evaluate the current methods to assess the uncertainty in RS-ET estimation



## 3.1 Identification and database search

The academic electronic databases Web of Science (WoS) and Scopus were searched (last access: 21.9.2021) using the combination of the three search terms: "evapotranspiration", "remote sensing", and "uncertainty", or their variants (Table 1). The term "transpiration" and "interception" were not used since they only represent components of total evaporation, which is
often referred to as evapotranspiration in remote sensing.

**Table 1: Search terms and variants. Search terms were combined using AND operator and variants were combined using OR operator. The asterisk * was used to include similar terms.**

| | Search terms combined by < AND > | | |
|---|---|---|---|
| **Variants combined by < OR >** | Evaporation | Remote sensing | Uncertainty |
| | Evapotranspiration | Remotely-sensed | Accuracy |
| | Latent heat | Remotely sensed | Data quality |
| | | Earth observation | Variability |
| | | Satellite* | Reliability |
| | | Global ** product | Evaluat* |
| | | Global ** data* | Validat* |

Since different authors might use different terms for satellite remote sensing, evaporation, and uncertainty in the title and
abstract, the variants of search terms were identified based on a set of 34 prior articles (Annex 1 in Supplementary Information). The search result was limited to a publication date from 2011 onward and then refined using the available filters of Scopus and WoS. Only English articles (>99% of results) that reported original research and were published in scientific peer-reviewed journals were considered. Review papers and conference proceedings were not included because they have different formats and provide no or few details of the methods used for uncertainty assessment. Gray literature was not included since its quality
was not assured by the scientific peer-reviewing system. Finally, we removed duplicates from the search result.

## 3.2 Relevance and eligibility screening

From the search result, we identified papers that attempted to assess the accuracy or uncertainty of one or more satellite remote sensing-based estimations, models, or datasets of terrestrial ET. The models of interest were diagnostic ET models as defined by Courault et al. (2005), Zhang et al. (2016), and Chen and Liu (2020). We screened the title and abstract of each paper to
identify the objective and methods of the research.

Due to the large number of articles in our search results, we applied a screening process using the ASReview software, a semi-automated screening system that incorporates an active learning classifier to predict relevance from text (van de Schoot et al., 2021) [Website: https://asreview.nl/]. In the ASReview screening interface, the user decides whether an article is relevant based on only the title and abstract, then ASReview shows the next articles in the order of relevance to the selected articles.
The screening interface does not show authors and affiliations to avoid subjective bias. Moreover, we could find relevant articles from the search result faster by stopping the screening process when predefined criteria are met. These "stopping" criteria are based on the number of articles and the efficiency of ASReview. Since ASReview can help find 95% of the eligible





studies after screening between only 8% to 33% of the studies (van de Schoot et al., 2021), we stopped screening when 100 (3%) irrelevant records had been found consecutively and at least 10% of the total records had been screened. By screening

titles and abstracts, we found 639 relevant articles from 3276 records (Figure 4).

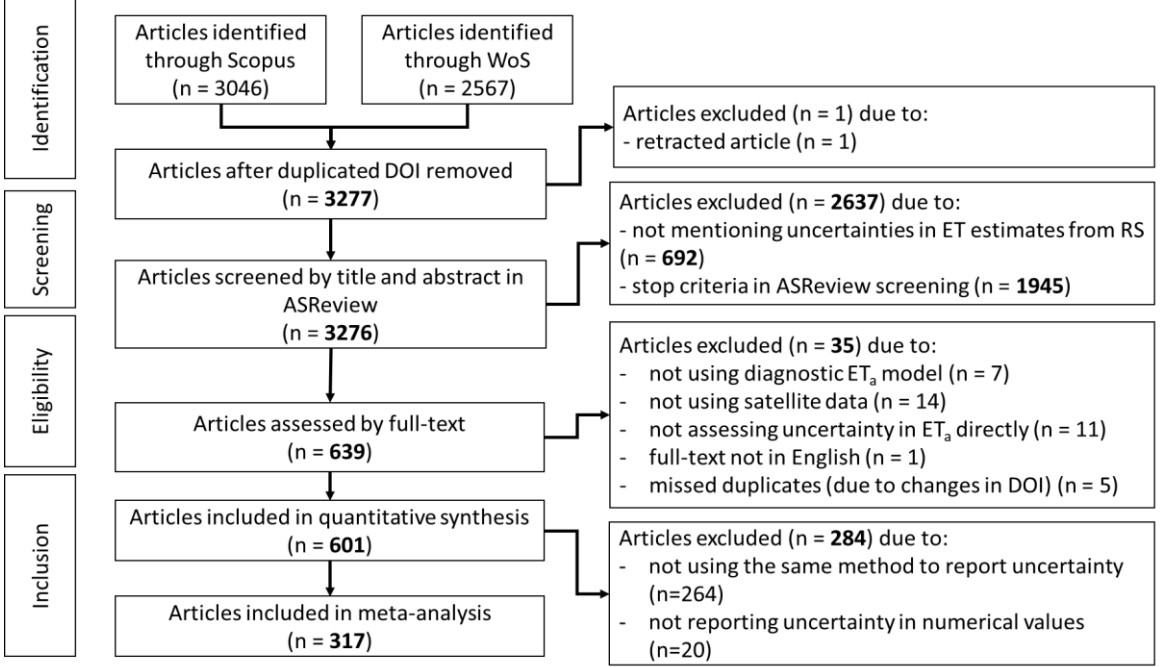

**Figure 4: Results of article selection from database search (identification), title and abstract screening (screening), and full-text assessment (eligibility).**

Even when screened with both abstract and title, without reading the full-text articles some non-eligible papers were still

included. For example, some studies used handheld radiometers, tower remote sensors, or airborne sensors, but only mentioned "remote sensing" in the abstract and title. Therefore, we assessed the eligibility of each paper by reading 639 full-text articles and finally included 601 articles in our review (Figure 4). For more information about the journals, authors, and year of publication, a brief bibliometric analysis of these 601 articles is provided in Annex 2 (Supplementary Information).

### 3.3 Article organization and analysis

From each included article, the items of information about the paper were recorded in a spreadsheet and were classified into categories based on methods, subject of research, and results (Table 2). The total number and percentage of research papers per category were then synthesized from the literature database, and the patterns and trends in assessing the uncertainty of RS-ET were discerned.

**Table 2: Categories and subcategories used to organize the included papers.**

| Categories group | Type of categories | Categories |
| --- | --- | --- |



| About the paper | Full reference details<br>Year<br>Journal title<br>Authors | |
|---|---|---|
| Objectives | Objective of the study | Model development<br>Model improvement<br>Model implementation<br>Product evaluation<br>Models evaluation |
| | Sources of uncertainties | Compound (Compared to reference)<br>Relative (Compared to other estimates)<br>Change of spatial support<br>Change of temporal support<br>Model parameterization<br>Input data<br>Gap filling |
| Methods | Types of approach | Sensitivity analysis<br>Uncertainty analysis<br>Validation<br>Inter-comparison<br>Others |
| | Uncertainty metrics | RMSE, bias, variance, … |
| | Types of reference | *In-situ* measurement (EC, lysimeter…)<br>Water balance |
| Scale of estimate | Temporal support | Sub-daily, daily, from 5 to 16 days, monthly, season, annual |
| | Spatial support | Less than 100 m, from 100 m to 500 m, from 500 m to 5km, from 5 km to 1°, more than 1°, basin, continent, global. |
| | Spatial coverage | Field, region, continent, global |

## 4 Review of methods for RS-ET uncertainty assessment

The selected articles assess uncertainty in RS-ET using 8 approaches: (1) validation, (2) intercomparison, (3) sensitivity analysis, (4) evaluation of input data, (5) uncertainty propagation, (6) three-cornered hat and triple collocation (TCH/TC), (7) physical consistency, (8) ensemble of estimates. Figure 5 shows the upset plot (Lex et al., 2014) of all reviewed articles by the approach of uncertainty assessment and the intersections of more than one approach. The majority of articles (532 out of 601) used a validation approach. Other approaches were much less frequently used and often in combination with validation, as shown by the number of intersections with 'Validation' (Figure 5).





**Figure 5: Approaches used in the reviewed articles (N=601). Intersections with less than 3 articles were excluded from the graph for improved presentation.**

Except for the validation and intercomparison approach, other approaches showed no increasing or a decreasing proportion in selected literature from 2011 to 2021 (Figure 6). Approaches other than validation and intercomparison have only been used by a small group of researchers and not applied widely or increasingly. This section will discuss the application of these approaches.



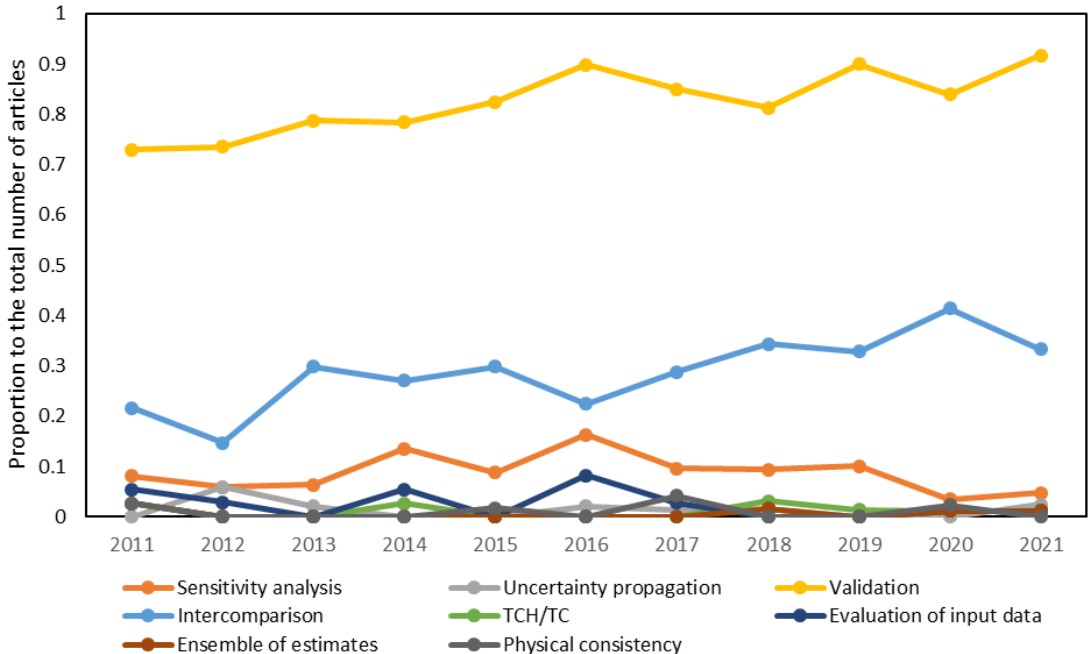

**Figure 6: The proportion of reviewed articles per year for each approach to assessing RS-ET uncertainties.**

## 4.1 Validation

In validation, RS-ET model results are compared to a 'reference' method that is considered by the researcher as the 'best' or most valid measure. The choice of the 'reference' method introduces subjectivity into the model evaluation (Melsen et al., 2019). In the case of RS-ET, three types of 'reference' are typically used: 1) *in-situ* measurements, 2) water balance (e.g., river basin, agricultural district), and 3) output from models run with ground-based input data. Almost all articles that used the validation approach considered an *in-situ* measurement as their reference (509 out of 532), while other types of reference data were much less considered.

### 4.1.1 Using *in-situ* measurement as the validation reference

Several *in-situ* methods have been developed to observe ET on the ground, including Eddy Covariance (EC), lysimeters, the Bowen ratio energy balance system (BREBS), etc. (Table S3 in Supplementary Information). These measurements are often considered the 'observation' or 'reference' to validate RS-ET. Among these, EC is the predominant method for validation and was considered in 386 out of 509 articles (Figure 7). Four factors explain the popularity of the EC method: 1) its relatively large network of stations, 2) long-term temporal coverage of flux towers, 3) open access of data (e.g., FLUXNET, EuroFlux, AmeriFlux, and OzFlux) and 4) direct measurement of water vapor concentration and vertical wind speed of the air parcels to calculate latent heat flux.





**Figure 7: Different *in-situ* methods used in reviewed articles (N = 509).**

Using *in-situ* methods for validation faces three main challenges: 1) the cost to set up and maintain measuring stations; 2) the mismatch between the source area of measurement and the spatial resolution of an RS-based estimate, and 3) errors in measurements and assumptions. For example, the cost of a complete EC system is about ten times the cost of a weather station with basic meteorological instruments. Although the EC method can be used to monitor other fluxes (e.g., carbon dioxide and nitrogen oxide), the high cost of the EC system still limits the number of sampling points and regions (Oliphant, 2012; FLUXNET, 2017). The low sampling density can be compensated with low-cost systems (Markwitz and Siebickeor, 2019) but at the expense of lower accuracy. In order to obtain validation data at global scale, EC networks need to be expanded in many regions (e.g., Africa, South Asia, the Middle East, and South America).

Spatial support of *in-situ* measurements often does not overlap with the pixel footprint of the RS images (i.e., the area the pixel value represents). The spatial support of *in-situ* measurements varies among methods, from 1 m$^2$ (micro-lysimetry) to a few km$^2$ depending on wind speed and wind direction (eddy covariance and scintillometry). For homogeneous pixels (with the



same geophysical and ecological characteristics), *in-situ* measurements can be representative of an entire pixel. However, when the pixel covers a large area, RS-ET validation frequently involves heterogeneous pixels. Therefore, multiple sites and upscaling methods are required to best aggregate site-specific to pixel-scale information (e.g., Liu et al., 2016; Li et al., 2018). Every *in-situ* measurement technique is subject to uncertainty and error. Even the most widely used technique, the EC flux

tower, has limitations in terms of measurement (10–20% error) and spatial support (Glenn et al., 2011; Wang et al., 2015). All methods have common sources of error and uncertainty, such as sensor response (detection limit), calibration error (sensor drift over time), noise (spurious random spikes in the signal from the sensor), and poor installation and maintenance (Allen et al., 2011a). Additionally, each method has specific sources of error and uncertainty due to its theoretical assumptions. For example, the EC method requires fully developed turbulent fluxes to ensure that the net vertical transfer of water vapor is

caused by eddies, and the area must be horizontal and uniform. Moreover, the lack of energy balance closure in EC measurements needs particular attention, since the gap can be up to 30% of available energy (Wilson et al., 2002; Vendrame et al., 2020, Bambach et al., 2022, Allen et al., 2011b). The problem is due to scale mismatch of energy balance components and unaccounted exchange fluxes on heterogenous landscapes (Foken, 2008).

Dealing with scale mismatch and uncertainty of reference *in-situ* measurements is challenging and there is no consistent

method in the reviewed literature. Some studies only mentioned these issues when discussing the validation result. The information about the spatial support and uncertainty of *in-situ* measurements is not always available to researchers if they acquire reference data from other sources. However, without reporting the spatial support and uncertainty of measurements, we might easily draw biased conclusions: when the validation results are good, we conclude that the model is good without questioning the quality of our reference, but when the results are not so good, we conclude that it is because of the imperfect

reference measurements that the model still is good. Hence, it is important to accompany validation results with the best knowledge about the uncertainty and scale mismatch of reference datasets.

### 4.1.2 Using the residual of the water balance as the validation reference

ET of an area can be estimated as the residual of the water balance (WB) when the inflow (e.g., precipitation, irrigation supply), change in storage, and outflows of water (e.g., runoff, water conveyance) of that area are known. This approach is mainly used

for assessment at a river basin scale. It assumes that the residual from the basin WB should be the total ET of the basin: $ET = P - Q - dS/dt$, where $P$ is precipitation, $Q$ is river discharge, and $dS/dt$ is the total change in basin water storage over the time period.

For long-term periods (e.g., years, decades), total water storage change (TWSC) over time ($dS/dt$) is assumed to be zero, such that ET estimates are then validated with only $P-Q$ (e.g., Vinukollu et al., 2011a). However, this assumption does not hold true

in many regions of the world where groundwater is being overexploited at an accelerated rate. For short-term periods (i.e., months), TWSC is often estimated from GRACE RS-based total water storage anomaly (TWSA) products. However, the GRACE TWSA products only cover the period from 2002 with a gap of 11 months from 2017 to 2018 due to the



discontinuation of the GRACE mission. Although techniques are being developed to reconstruct this gap in the GRACE time series (e.g., Yang et al., 2021), the accuracy of *dS/dt* estimates from GRACE is still less known.

The uncertainty in ET estimated by this approach depends on the choice and data quality of other variables (e.g., precipitation and river discharge) in the WB (Senay et al., 2011). Lehman et al. (2022) have compared the residual calculated from 1694 combinations of *P*, *Q*, and *ET* datasets with *dS/dt* derived from GRACE and found that none of these combinations can close the WB in all tested basins. They also suggested that using some combinations of *P*, *Q*, and *ET* datasets cancels out their errors in the GRACE-based WB. Because of the errors in the *P*, *Q*, and *dS/dt* components, studies that use WB-derived ET as a

reference to validate ET without accounting for uncertainties in the *P*, *Q*, and *dS/dt* components risk biased conclusions.

In order to account for errors in *P*, *Q*, and *dS/dt*, some researchers have tried to use multiple datasets (e.g., Weerasinghe et al., 2020). Recently, Schoups and Nasseri (2021) proposed treating uncertainties in datasets as unknown random variables. Instead of using the WB to determine these uncertainties, they estimated ET (and other water fluxes) by combining WB constraints and uncertainty estimation into a comprehensive probabilistic model. Although only applicable for river basins where GRACE

resolution is suitable, this could be a good direction for future research on these water fluxes.

## 4.2 Intercomparison

Intercomparison is the second most widely used method (195 out of 601 studies). In intercomparison, the RS-ET estimates from multiple models are compared without assuming a superior one. This approach is mainly used to evaluate the relative uncertainty of a model compared to others (155 out of 195 studies). Intercomparison has also been used to evaluate other

sources of uncertainty. For example, uncertainty from a change of spatial support can be evaluated by comparing model outputs using different input upscaling methods (e.g., Ershadi et al., 2013; Sharma et al., 2016). Intercomparison has also been used to evaluate uncertainty due to the choice of input datasets (e.g., Long et al., 2011; Wang et al., 2016; Badgley et al., 2015).

Since the RS-ET datasets have both temporal and spatial dimensions, comparing RS-ET models or products is usually done by aggregating over one or two dimensions (i.e., resampling to a lower resolution). The simplest method of intercomparison

involves aggregating ET estimates both temporally and spatially into one value (e.g., global annually averaged ET) and then comparing this value from different models or products (e.g., Mueller et al., 2013; Pan et al., 2020). Other methods of intercomparison involve comparing time series of spatially aggregated ET (e.g., monthly basin-scale ET). Aggregating over one of the two spatial dimensions is sometimes applied (e.g., Pan et al., 2020; Chen et al., 2019). The time series can also be aggregated by land cover classes (e.g., Weerasinghe et al., 2020) or climate zones (e.g., Trambauer et al., 2013), describing

how RS-ET uncertainty varies in different conditions. For spatial intercomparison, temporally aggregated RS-ET maps can be compared visually (e.g., Weerasinghe et al., 2020) or by using simple map algebra (e.g., Jung et al., 2019). Only a few studies have applied metrics to evaluate the spatial similarity between two datasets, such as the Spatial Efficiency metric (SPAEF) (Stisen et al., 2021; Jung et al., 2019) and the degree correlation measure of spherical harmonic coefficients (López et al., 2017). None of these methods can characterize uncertainty in RS-ET fully, thus, combining them would provide a more

comprehensive intercomparison.



### 4.3 Sensitivity analysis and uncertainty propagation

Sensitivity analysis (SA) is the third most used approach in the reviewed literature, but is only applied a in a small proportion of the reviewed studies (53 out of 601 articles). Out of these, only 6 studies applied global sensitivity analysis (GSA). Sobol's (2001) method was applied to the parameters of the MODIS16 algorithm (Zhang et al., 2019), the TSEB model (Burchard-

Levine et al., 2020), and three RS-ET models (PT-DTsR, MODIS16 algorithm, and PML) (Cao et al., 2021). This method was also applied to input variables of RS-ET models alone (e.g., Gomis-Cebolla et al., 2019). Elhag (2016) applied a similar variance-based sensitivity measure for the SEBS model but did not refer to Sobol's method. The Extended Fourier Amplitude Sensitivity Test (EFAST) has also been applied for GSA (García et al., 2013). This limited number of studies shows the application of GSA onto RS-ET models has been under-researched during the last decade, despite the importance of GSA in

environmental modeling (Saltelli et al., 2021).

The majority of articles that applied sensitivity analysis (47 out of 53) did not mention or apply a GSA method and thus, were considered to be local sensitivity analysis (LSA). In most of these studies, LSA was done by changing one parameter at a time (One-at-A-Time or OAT) and calculating the ratio of change in ET over change in parameter (e.g., Long et al., 2011). In the reviewed articles, the OAT method has been implemented differently in terms of three factors: 1) the selection of parameters

for LSA according to their importance judged by the researchers, 2) the range of values over which parameters are allowed to vary, and 3) the calculation of sensitivity for specific land covers. This suggests that LSA is influenced by the subjectivity of the researchers.

Only 7 out of 601 articles applied the uncertainty propagation approach, mainly MCM, to evaluate uncertainty in RS-ET. The limited application of uncertainty propagation can be attributed to its complexity and computational demand. Sensitivity

analysis and uncertainty propagation are ideally carried out in tandem (Crosetto et al., 2001; Saltelli et al., 2019), but only 4 out of 7 articles combined these approaches. The uncertainty propagation approach was also used for investigations beyond uncertainty quantification. For example, Talsma et al. (2018) used MCM to determine the uncertainties in ET partitioning (i.e., soil evaporation, interception, and transpiration) in 3 RS-ET models (MOD16, PT-JPL, and GLEAM) due to the relative uncertainty in the key variables.

In the reviewed studies, uncertainty propagation was done only at one or a few fixed locations by assuming the probability distribution of the input variables, then simulating a range of ET values at these locations. This approach is computationally inexpensive but does not fully characterize uncertainties in a spatial field of ET. To fully quantify uncertainty in a scene, Cawse-Nicholson et al. (2020) introduced a method based on MCM and spatial-statistical models (Cressie, 1993). With this method, the probability distribution of ET per pixel in a satellite scene can be quantified and presented as percentile maps.

This distribution was almost always non-Gaussian for all pixels in ET scenes, which means simple linear error propagation is not possible (Cawse-Nicholson et al., 2020). Future studies of RS-ET would benefit from the development of new methods to quantify uncertainty spatially.





## 4.4 Evaluation of input data

The uncertainties of key input datasets are sometimes evaluated by researchers in studies that assess uncertainty in RS-ET
without explicitly being propagated to model outputs. This approach ranked fourth in the number of articles with 12 out of
601. The key input datasets considered by researchers include air temperature, incoming shortwave radiation, incoming
longwave radiation, wind speed, and land surface temperature (e.g., Vinukollu et al., 2011; Pardo et al., 2014; Peng et al.,
2016; Li et al., 2017). Input datasets were evaluated through validation with their *in-situ* counterpart. Although other input
datasets like Vegetation Indices are also important in RS-ET models, the *in-situ* measurements of these are often not available
for evaluation (Vinukollu et al., 2011). Some of the forcing datasets of RS-ET models are not remotely sensed data but are
products from atmospheric data assimilation systems (e.g., Global Land Data Assimilation System (GLDAS) and ECMWF
atmospheric reanalysis (ERA)), which are sometimes provided with uncertainty estimates from data providers. Evaluating the
input data provides crucial *a priori* information for propagating uncertainty to ET estimates. Furthermore, even if uncertainty
propagation is not conducted, these assessments can help to identify sources of uncertainty in RS-ET; as the saying goes,
"garbage in, garbage out".

## 4.5 Triple Collocation and Three-cornered hat method

The Three-cornered hat method (TCH) (Premoli et Tavella, 1994) and triple collocation (TC) (Stoffelen, 1998; McColl et al.,
2014) are related to the intercomparison approach in the sense that these techniques assess the relative uncertainty of three
datasets without assuming one is the best. Therefore, these techniques are useful when there lacks a high-quality reference
dataset. Both TC and TCH methods require a set of three datasets with the assumption that their errors are independent (Sjoberg
et al., 2021). The difference between TCH and TC is that TC can only be used to assess uncertainties of uncorrelated datasets,
while TCH can be used when there are correlations with proper constraints (Xu et al., 2019; Sjoberg et al., 2021). However,
to date few studies have evaluated uncertainties in RS-ET using TC (Miralles et al., 2011b; Barraza Bernadas et al., 2017;
Khan et al., 2018; and Kibria et al., 2021) and TCH (Long et al., 2014; Xu et al., 2019; and He et al., 2020). The proportion of
studies that used these methods is less than 5% of the total reviewed articles and is not increasing (Figure 6). This low adoption
might be attributed to the limitations of these methods: 1) the lack of information about biases and only estimation of random
errors (e.g., RMSE, standard deviation, or variances), 2) the required conditions to achieve reliable error estimates (large
samples, similar scales and magnitudes of errors between datasets) (Sjoberg et al., 2021), and 3) the reliability of TCH as an
alternative to direct validation (Wu et al., 2019b).

## 4.6 Physical consistency


Physical consistency can be understood as the plausibility that an ET estimate is consistent with the physical conditions or
characteristics of the area it represents. Consistency check or physical validation was proposed by Zeng et al. (2015) as the
final step in a general validation process for big remote sensing datasets. When there is limited reference data and ground-



based measurements, physical validation is critical to assess the quality of data products (Blatchford et al., 2020). Although
physical validation does not quantify uncertainty using metrics, it provides an evaluation of the data quality. This is useful to
identify the regions and conditions in which RS estimates are more uncertain and where more effort in direct validation
approaches is required.

Only 6 studies in the selected literature have attempted to quantify this plausibility (Figure 6), but they defined physical
consistency differently. For example, Rwasoka et al. (2011) used FAO Penman-Monteith potential ET estimates as a threshold
to decide whether ET estimates from the SEBS model were physically inconsistent. Blatchford et al. (2020) used the *ET/P*
ratio and water availability (*P-Q*) to evaluate the physical consistency of the WaPOR ET product. López et al., (2017)
developed a technique to assess the hydrological consistency of ET by transforming both ET and P data into spherical
harmonics and then using spherical harmonic coefficients to calculate the degree correlation. These studies are not the same
as validating RS-ET with *P-Q* or *P-Q-dS/dt* as discussed previously, since these residuals were not considered the best
reference of ET.

Another method to assess physical plausibility without explicit water balance is through the Budyko curve. The Budyko curve
describes the semi-empirical relationship between long-term ET and its limiting factors, i.e., precipitation and potential ET
(PET), for river basins (Budyko, 1974). Koppa et al., (2017) validated the physical consistency of ET by calculating the RMSE
of the Euclidean distance between the data points and the Budyko curve in *ET/P* and *PET/P* space. Weerasinghe et al., (2020)
simply calculated the mean difference (bias) between RS-ET and Budyko-derived ET to evaluate which RS-ET product
exceeds the energy and water limit defined by the Budyko curve. They also noticed that if a data point does not align with the
Budyko curve, it might also mean that the ET of the basin exceeds the water or energy limit, for example, due to human
activities. Therefore, the interpretation of physical plausibility needs to consider the actual knowledge about water resources
in the basin, instead of focusing only on model generated numbers.

**4.7 Ensemble of RS-ET estimates**

Intercomparison studies sometimes lead to ensemble-mean products of all available products, on the basis of the assumption
that no model performs best, so an ensemble of them would be preferable (Bhattarai et al., 2019; Elnashar et al., 2021). This
approach has been used the least in the reviewed articles (Figure 6). Some researchers have evaluated the uncertainty in an
ensemble (a set) of RS-ET estimates from different models by calculating the average and range of all members in the ensemble
(Vinukollu et al., 2011b; Elnashar et al., 2021; Guo et al., 2020). This approach is the same as the multi-model ensembles in
climate modeling. The model structural uncertainty can only be quantified if independent models are sampled from the entire
possible model space and avoid the over-representation of one model structure (Abramowitz and Gupta, 2008). For example,
Vinukollu et al. (2011b) selected three RS-ET models, namely SEBS (Su, 2002), PM-Mu or MODIS16 (Mu et al., 2007), PT-
Fi or PT-JPL (Fisher et al., 2008), which are based on distinct equations used to estimate ET.
The ensemble approach provides uncertainties of the ensemble but not each individual member of the ensemble. Thus, some
studies went further by merging the datasets of the ensemble and calculating the difference between this merged dataset with



each ensemble member (Baik et al., 2018; Elnashar et al., 2021). If simply averaging all the ET products, the bias of different models can be canceled in regions where they perform differently but accumulated in regions where they perform in the same manner. Hence, the ensemble products may arguably produce better estimation in some areas, but not a better understanding
of the physical processes and drivers needed to improve RS-ET (Zhang et al., 2016). Therefore, it is considered more useful to use the range of the ensemble to identify the outlier data products or the uncertainty of all data products.

## 5 Context of RS-ET uncertainty assessment

The context in which the uncertainty of RS-ET is assessed determines which method is selected and how it is applied. This context includes the objective of the RS-ET estimates, the spatial and temporal support at which ET is assessed, geographic
location, and the availability of reference datasets. This section describes the context in which 601 reviewed articles assessed uncertainties in RS-ET.

### 5.1 Research objectives

The review shows that uncertainties in RS-ET estimates were assessed at all stages, from developing a new model to evaluating its data product. Uncertainty in RS-ET was assessed in the context of model implementation (33% of reviewed articles), model
development (12% of all reviewed articles), model improvement (17%), model evaluation (19%), and product evaluation (17%) (Figure 8). Here, model implementation means that a pre-existing model was applied to new case studies or to achieve some specific research objective without considerable modification or further development of the model. The prominence of model implementation as the main objective in the reviewed articles could be due to a perceived need to assess the uncertainty of RS-ET estimates for each application despite previous validation. This is an important attitude in the research community
since it helps to provide feedback on appropriate application, and improvement of RS-ET models. Therefore, studies in the context of model implementation should not be overlooked.



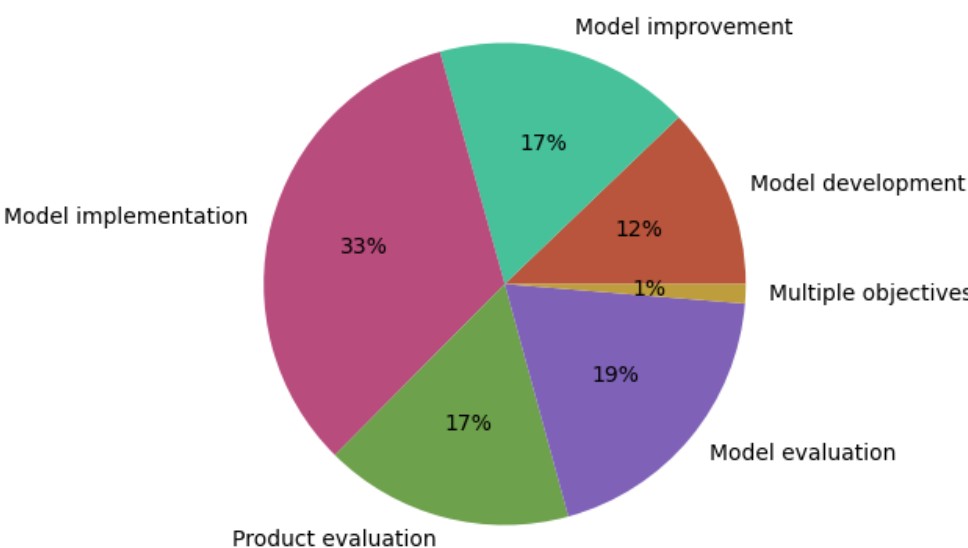

**Figure 8: Research objective of the reviewed articles (N=601).**

## 5.2 Sources of uncertainty evaluated

The reviewed articles evaluated all sources of uncertainty as categorized in the theoretical framework (Figure 3), with strong
focus on compound uncertainty. Figure 9 shows that the majority (360 out of 601) of reviewed articles assess only compound
uncertainty without disaggregating into other sources. The second largest set of articles assessed both compound uncertainty
and the relative uncertainty of RS-ET estimates. Other sources of uncertainty are remarkably less evaluated in the selected
literature. According to the number of articles in each set (Figure 9), the level of interest in different sources of uncertainty can

be ranked as follows: compound, relative, input data, model parameterization, change of spatial support, change of temporal
support, and finally gap filling. This does not necessarily show the ranking of importance of the uncertainty sources, but rather
the availability of methods and data needed to assess them.



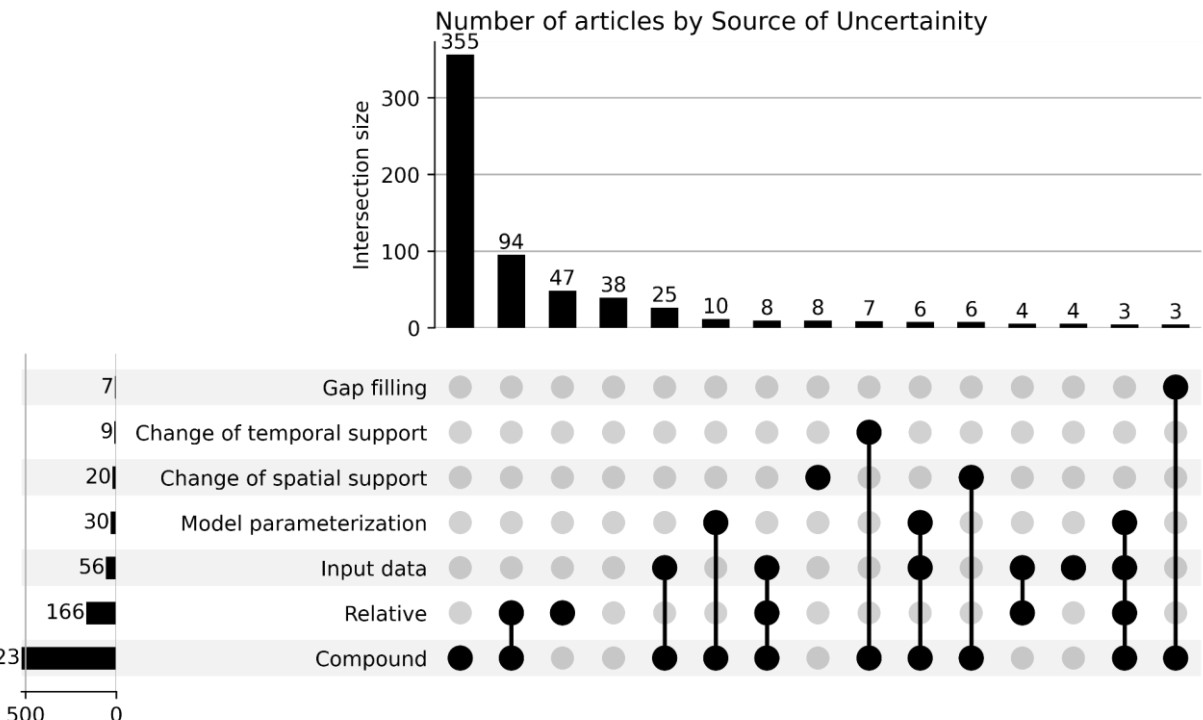

**Figure 9: The source of uncertainty assessed in reviewed articles (N=601). Intersection with less than 3 articles were excluded from**

**the graph for improved presentation.**

### 5.3 Spatial and temporal support

Uncertainties in RS-ET estimates are specific for different spatial and temporal supports. The reviewed studies evaluated RS-ET uncertainties at spatial supports ranging from less than 100 m up to global, and temporal support ranging from sub-daily to annual (Figure 10). Most studies evaluated RS-ET uncertainties at spatial supports of 500 m to 5 km (244 out of 601) and

less than 100 m (164 out of 601). This can be attributed to the availability of RS datasets that are widely used to estimate ET, such as MODIS (250 m to 1 km) and Landsat (30 m to 100 m). In the case of validation, the spatial support of uncertainty assessment was determined by the spatial support of the ground truth reference. For temporal support, uncertainty was mostly evaluated by daily ET (328 out of 601), although RS datasets provide observations at the time of satellite overpass with a temporal resolution of 5-16 days. Thus, RS-ET models must estimate instantaneous ET and then upscale to daily ET, which is

more useful in ET applications (e.g., irrigation monitoring). This shows that the temporal support of uncertainty assessment is driven more by practical needs and less by the availability of datasets.



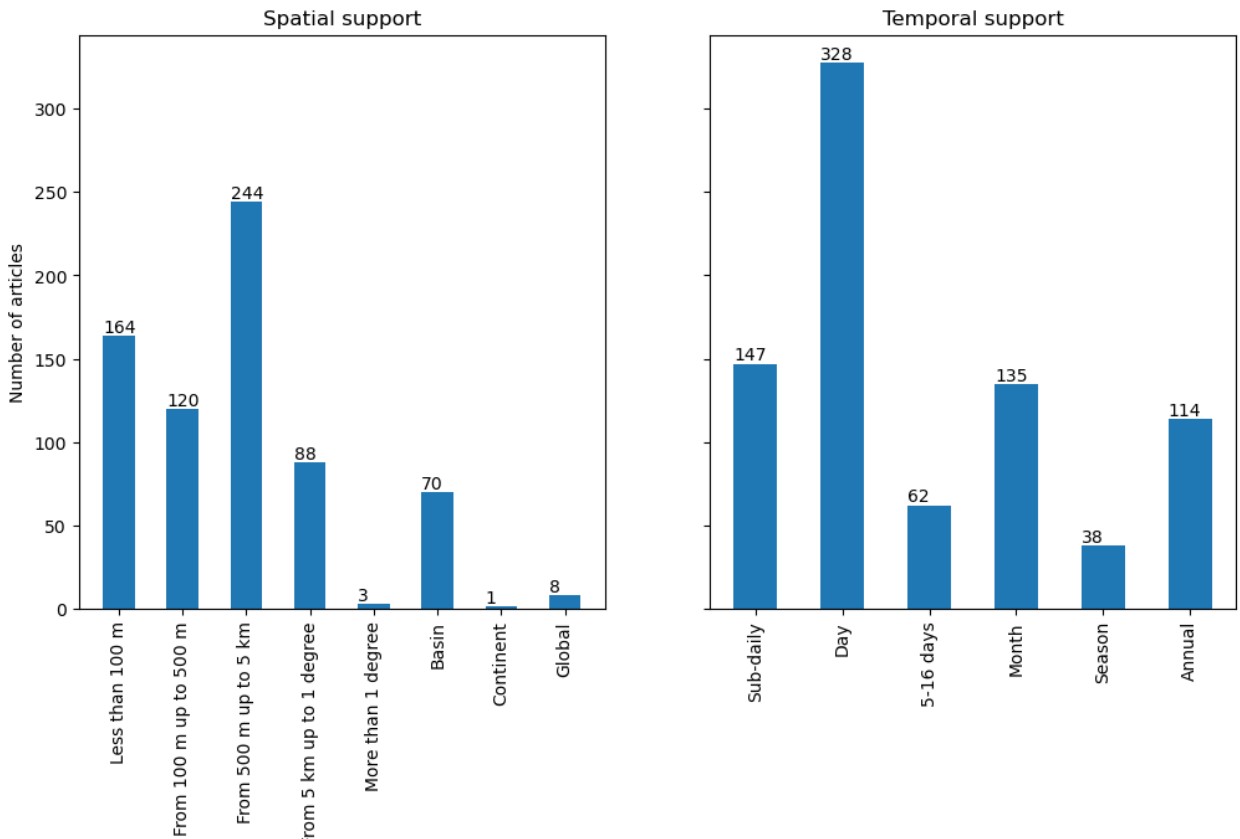

**Figure 10: Number of articles per range of spatial and temporal support at which uncertainty in RS-ET was assessed (total number of articles N=601).**

## 5.4 Geographical distribution

Assessment of RS-ET uncertainties is not evenly distributed over the globe. The number of articles per country where uncertainties in RS-ET were assessed is shown in Figure 11. Each article was tagged by the country where the sites of study is located. The highest number of articles assessed ET in China. Because the most common approach is validation and the most common reference used is EC measurements, ET was mainly assessed where there are EC stations (i.e., AmeriFlux, AsiaFlux, ChinaFlux, OzFlux, EuroFlux, FLUXNET). Even when the studies aimed to validate RS-ET globally, the estimated uncertainty is not universal since these networks do not cover many regions. These studies were also included in Figure 11.



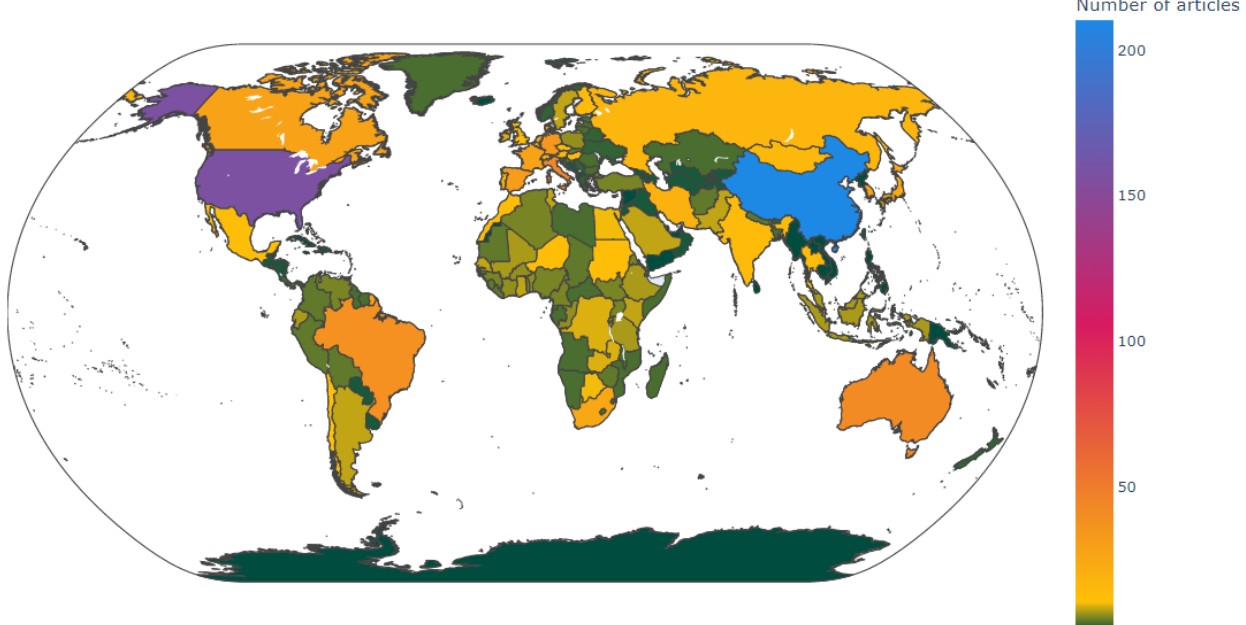

**Figure 11: Number of articles per country where uncertainties in RS-ET were assessed.**

Based on its popularity, EC can be considered the de facto standard ET estimation approach for validation of RS-ET. However, this popularity is mainly driven by the number of publications in countries where EC towers are more densely distributed (e.g., China and the United States of America). In countries where there are very few or no EC towers available, the most common reference used for validation of RS-ET is the water balance method (Figure 12). In a few countries in North Africa and the Middle East, the most common method is to use the FAO-56 method (Allen et al., 1998) in combination with crop coefficients to estimate ground-based references for validation (e.g., Egypt and Iran).



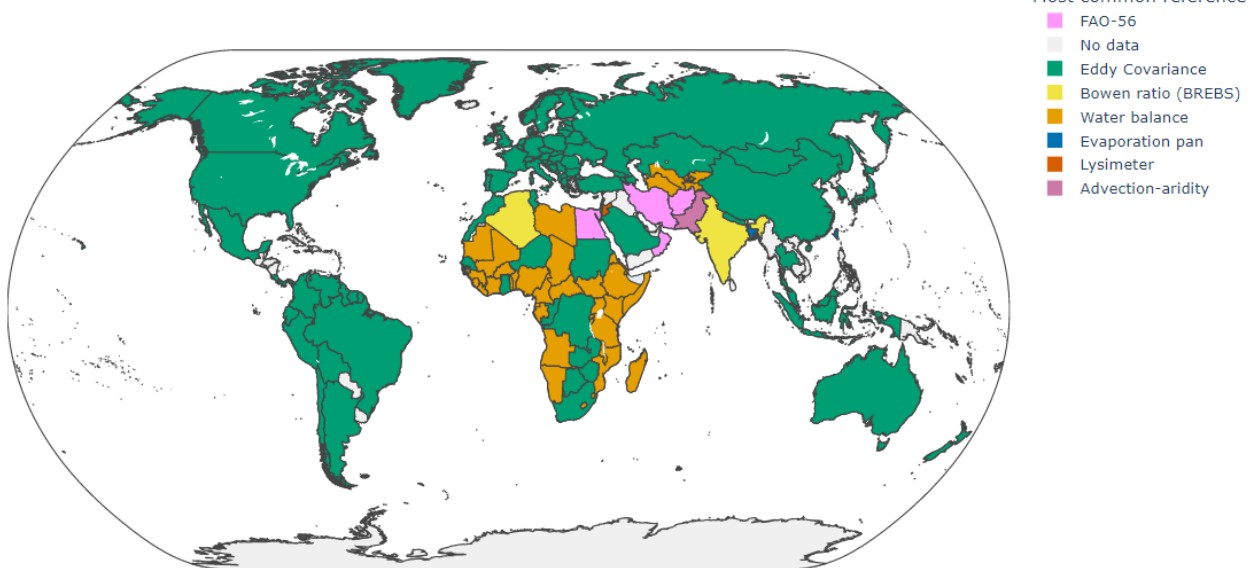


**Figure 12: The most common reference used for validation of RS-ET per country.**

# 6 Results of RS-ET uncertainty assessment

## 6.1 Uses of uncertainty metrics

The reviewed articles that assess uncertainty in RS-ET mainly report accuracy (RMSE), bias (mean error), and the goodness-of-fit with a reference dataset ($R^2$) (Figure 13). Although quantifiable uncertainty in measurement is theoretically represented as a probability distribution, this has rarely been done in the literature. The reviewed studies used a wide range of metrics to report their uncertainty assessment (33 metrics). Most studies used three metrics, while some used up to twelve. Larger number of metrics provide more description of uncertainty, but some metrics might be challenging to interpret.




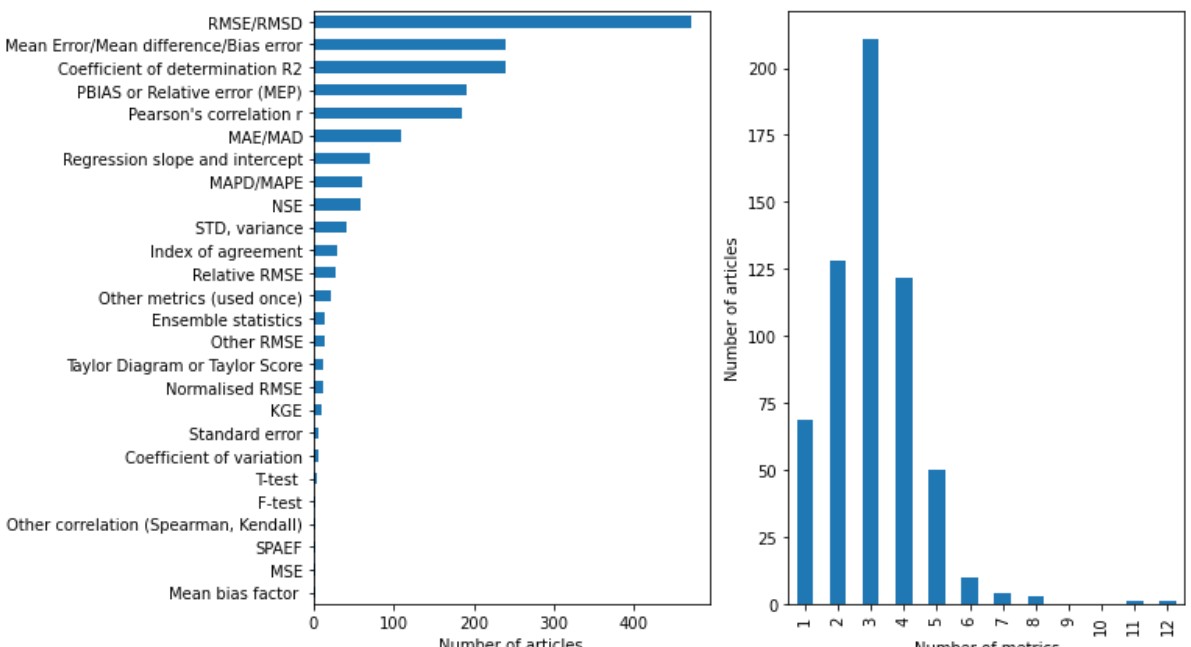

**Figure 13: Number of studies per choice of metric to report uncertainty and the number of metrics used.**

Root Mean Square Error (RMSE) is the most widely used metric in the reviewed articles (467 out of 601 articles). Metrics related to RMSE include normalized RMSE (normalized by standard deviation) and relative RMSE (as a percentage of mean ET). Very few studies (13 articles) used modified RMSE to report more robust results and few consider random error and systematic error, such as robust RMSE (Bisquert et al., 2016), systematic and unsystematic RMSE (Yebra et al., 2013), biased

and unbiased RMSE (Martens et al. 2017). RMSE has the unit of the estimates, so it can be expressed in mm for ET or Wm$^{-2}$ for latent heat flux. Therefore, to compare reported RMSE between different studies, unit conversion is needed.

Inconsistent use of metrics such as $R^2$ might cause misinterpretation of results, especially when comparing studies. For example, the second most used evaluation metric was referred to using many names including mean error, mean difference, bias error, or bias. Meanwhile, the coefficient of determination ($R^2$) has the opposite issue, in which the same term was used

with different formulas. $R^2$ is a measure of goodness-of-fit for regression models. There are at least 8 formulas for $R^2$ in the literature (Kvålseth, 1985), but only one formula can be used for any type of model fitting (i.e., $R_1^2$ in Kvålseth, 1985). Since many studies did not report which formula they used, we did not distinguish between different $R^2$ formulas in Figure 13. Nevertheless, we observed that at least four different formulas of $R^2$ were used in the reviewed articles including the squared coefficient of correlation (Table S4 in Supplementary Information).

No matter which metrics are used, the validation metrics that compare estimate with reference only represent actual error if the reference is the absolute truth. This is never the case because *in-situ* measurements and upscaling methods are never perfect. Wu et al. (2019a) suggested that validation should be performed in conjunction with uncertainty associated with *in-situ* measurements and the statistical significance of performance metrics.



## 6.2 Synthesizing reported RS-ET uncertainty from reviewed studies

Although there are a large number of papers assessing uncertainty of RS-ET data, only a few attempted to synthesize their results. For example, a review by Karimi and Bastiaanssen (2015) used meta-analysis (i.e., using statistical methods to synthesize results of independent studies) to estimate the probability density function of mean absolute percentage error (MAPE) in 46 studies that validate RS-ET estimates in seasonal cycles. Kalma et al. (2008) summarized the relative error and RMSE of RS-ET reported in 30 studies. These syntheses are limited in number of studies (<50) and the selection of studies

were not systematic. Another limitation of synthesizing these results is that the selected studies used different validation data and field instruments, which do not have equivalent spatial support and accuracy.

Synthesizing results of the reviewed articles in this study will provide a useful reference for future studies to evaluate the results of RS-ET uncertainty assessment. For a meta-analysis, selected studies should use the same validation data and report the same metric, thus, we selected the most used validation data and metric. Since the majority of studies used EC flux towers

and RMSE to report uncertainty (337 out of 601), we selected these studies for meta-analysis of reported RS-ET uncertainty. From 337 articles, 318 articles that reported RMSE of RS-ET from validation with EC flux tower were included. The remainder were excluded because the RMSE was not reported in figures with extractable values (Figure 4). RMSE values in units other than mm/day were converted to mm/day assuming constant rate of ET over the temporal support. For example, 365 mm/year was converted to 1 mm/day and 0.1 mm/hour was converted to 2.4 mm/day.

The reported RMSE values for daily ET (N = 2,407) range from 0.01 to 6.65 mm/day with the mean value of 1.12 mm/day (Table 4). When converting RMSE values from the reported unit to a common unit of mm/day, the mean RMSE is the highest for validation of instantaneous RS-ET (2.81 mm/day) and the lowest for monthly (0.50 mm/day). In general, studies with larger temporal support of validation have lower mean RMSE in mm/day. For the validation at temporal support of 3-hour, 10-day, and week, less can be concluded due to small number of studies and records. Overall, the decrease of RMSE with increasing

temporal support is due to the averaging and corrective effect of temporal upscaling. Therefore, improving temporal upscaling and gap-filling methods are crucial for reducing uncertainty in RS-ET estimates.

**Table 4: Descriptive statistics of reported RMSE values (in mm/day) in reviewed articles (N=317) with validation of RS-ET with EC flux towers.**

| Temporal support | Number of records | median | mean | standard deviation | min | 25th percentile | 75th percentile | max |
|---|---|---|---|---|---|---|---|---|
| instantaneous | 680 | 2.64 | 2.81 | 1.47 | 0.20 | 1.65 | 3.62 | 8.63 |
| 30-min | 130 | 1.58 | 1.66 | 0.80 | 0.16 | 0.99 | 2.16 | 4.13 |
| hour | 119 | 0.56 | 0.88 | 0.97 | 0.16 | 0.35 | 1.14 | 5.99 |
| 3-hour | 18 | 2.48 | 2.58 | 0.80 | 1.44 | 1.92 | 3.11 | 4.54 |
| day | 2407 | 0.94 | 1.12 | 0.71 | 0.01 | 0.71 | 1.27 | 6.65 |
| week | 13 | 0.63 | 0.49 | 0.29 | 0.02 | 0.17 | 0.71 | 0.81 |
| 8-day | 454 | 0.75 | 0.88 | 0.48 | 0.04 | 0.56 | 1.11 | 3.40 |



| | | | | | | | |
|---|---|---|---|---|---|---|---|
| **10-day** | 15 | 0.90 | 1.04 | 0.46 | 0.40 | 0.80 | 1.16 | 2.20 |
| **16-day** | 53 | 0.49 | 0.50 | 0.14 | 0.22 | 0.40 | 0.62 | 0.89 |
| **month** | 469 | 0.60 | 0.74 | 0.53 | 0.03 | 0.38 | 0.97 | 4.96 |
| **year** | 71 | 0.83 | 0.80 | 0.31 | 0.15 | 0.61 | 1.02 | 1.49 |
| **overall** | 4429 | 0.97 | 1.31 | 1.07 | 0.01 | 0.67 | 1.50 | 8.63 |

Figure 14 shows that very high RMSE values were mainly from validation approaches that used a single EC site. Validation based on a large number of EC sites are more likely to result in lower RMSE values. As shown in this study, the distribution of validation sites is concentrated in regions where EC flux towers are available. The results of the validation, thus, are not necessarily transferable to other areas because the control factors of ET and uncertainty in their estimates are not the same globally (Zhang et al., 2016). Therefore, when interpreting the uncertainty of RS-ET based on validation, we should consider

the validation metrics at each site individually and the variation of these metrics among all locations.

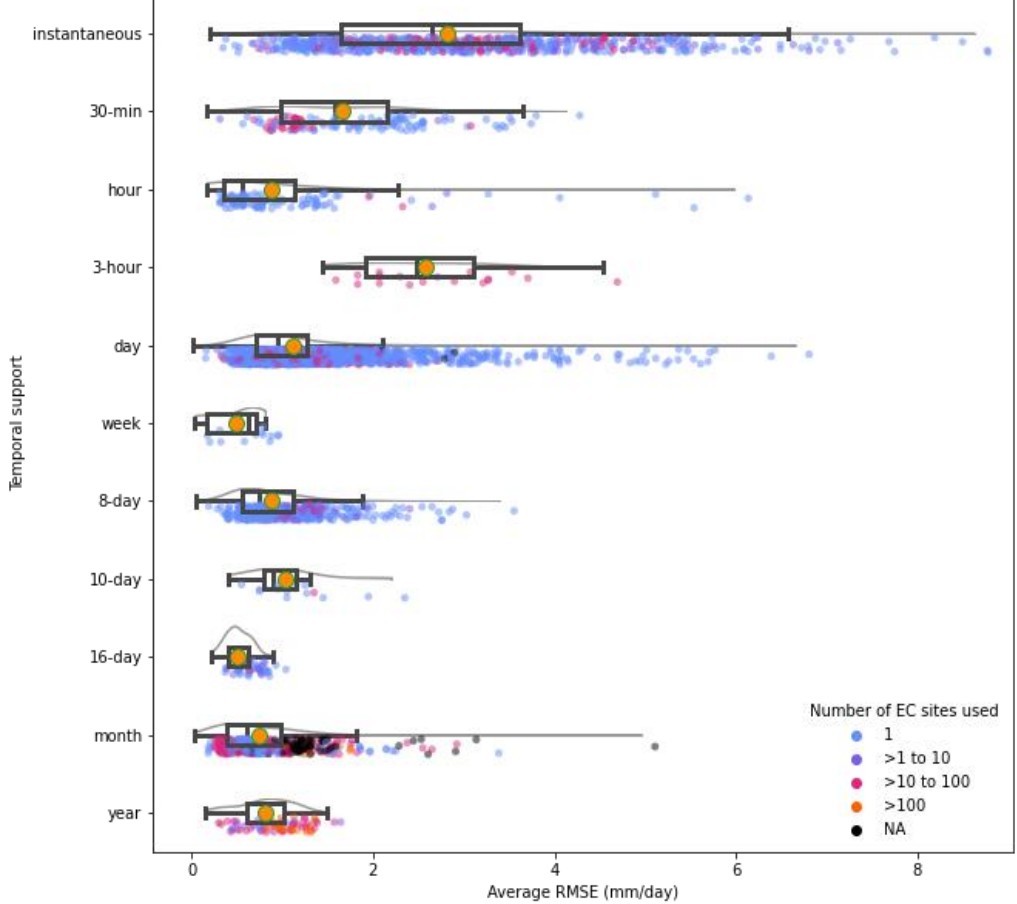



**Figure 14: RMSE (mm/day) of RS-ET based on validation with Eddy Covariance observations in reviewed articles (N= 317). Each dot represents one RMSE value reported in articles. The dot color shows the number of EC sites used in validation.**

The large range of RMSE obtained from the meta-analysis can be explained by the diversity of reviewed studies in terms of
models, resampling methods, and validation context (e.g., temporal scales, land cover, climate, amount of data). For example, some studies validate RS-ET estimates from global products, while others validate RS-ET estimates from models that were calibrated to reduce RMSE. Moreover, many studies reported RMSE of latent heat flux (in Wm-2 or MJm-2day-1) averaged from estimates at the time of satellite overpass. The accuracy of RS-ET varies at different times of the day due to weather condition, and is, thus, not representative of the entire day. We converted these values to mm/day (Table 4) only for comparison
between different temporal supports.

## 7. Summary

This paper identifies and appraises methods for uncertainty assessment of RS-ET estimates. We applied the systematic quantitative literature review (SQLR) approach to identify the number of original research papers in distinct categories of studies, which includes scale, region, source of uncertainty, and method. The majority of reviewed articles assess uncertainty
in RS-ET estimates by validation against EC measurements. Making use of existing EC networks is important for global validation of RS-ET estimates. However, there is still a gap *in-situ* data for global validation, as most are concentrated in North America, East Asia, and Europe. Moreover, the challenges in energy balance closure and scale mismatch persists through  the reviewed studies. In regions where *in-situ* measurements are limited, most studies used the residual of the water balance as a reference for validation.

The uncertainties of RS-ET are not easy to quantify and characterize ina single study. Future research should combine local and global evaluation efforts. Combining multiple approaches for uncertainty assessment of spatiotemporal RS-ET data is recommended due to limitations of validation datasets. Approaches other than direct validation includes intercomparison, sensitivity analysis, uncertainty propagation, physical consistency check, evaluation of input, triple collocation, and the ensemble of estimates. Both sensitivity analysis and uncertainty propagation approaches were shown to be useful for the
advancement of RS-ET techniques by identifying and quantifying the sources of uncertainty. However, applying uncertainty and sensitivity analysis approaches to remote sensing retrieval models over a large spatial extent is computationally demanding. We showed in our review that there are very few studies that applied sensitivity analysis and uncertainty propagation techniques for RS-ET estimates and that their methods are limited to less computationally demanding options. This impedes the ability to assess a detailed spatiotemporal distribution of RS-ET uncertainty. Therefore, future research on
uncertainty in RS-ET estimates needs to develop and apply more advanced sensitivity analysis and uncertainty propagation methods.



As the majority of reviewed articles used validation, we provide the range of uncertainty in RS-ET based on a meta-analysis of 317 articles that reported uncertainty in RS-ET as the RMSE compared to EC observations. The RMSE range reported in our study can be used in future research to compare and evaluate results with the majority of validation studies. RMSE varies a lot among validation sites and temporal supports. Moreover, validation with more than one station reported lower value and smaller variation of RMSE than validation at single site. Therefore, validation metrics only reflect uncertainty of RS-ET at specific locations. For future researches that validate RS-ET estimates with *in-situ* methods, we provide specific recommendations:

- The uncertainty of the reference datasets, including correction for surface energy balance closure, should be evaluated and reported.
- Temporal and spatial support of reference datasets should be matched as much as possible to that of the RS-ET estimates.
- Multiple metrics should be used to better characterize uncertainties in RS-ET estimates.
- The four common metrics (RMSE, bias/mean error, correlation coefficient, coefficient of determination), mean ET, the number of data points, and statistical significance test should be reported.
- Validation results should be reported at various levels of spatial and temporal support and at multiple locations with different characteristics
- The statistical significance of validation metrics should be tested and the number of data points used should be reported.

Since uncertainty in RS-ET is an attribute of any spatiotemporal dataset, the remaining challenge is to characterize uncertainty spatially and temporally. This means not only quantifying the overall expected errors of the dataset but also identifying where and when high uncertainty is most likely to occur. Several studies in remote sensing have attempted to provide spatially explicit uncertainty of thematic classification (e.g. land cover, soil type), such as the studies listed by Woodcock (2002). For quantitative remote sensing, which is the mapping of continuous variables like ET, methods to characterize spatially-explicit uncertainty are desirable but have not been developed as much as thematic mapping (e.g. pattern recognition, feature extraction). Therefore, we recommend that methods that evaluate spatiotemporal uncertainty need to be developed and applied to RS-ET datasets.

**Data availability**

The systematic categorization and analysis of the reviewed articles are available at https://doi.org/10.4121/797dcaff-56e3-45ae-a931-f6f4a3135d26.v1

The reported RMSE data from the reviewed articles that used Eddy Covariance to validate Remote Sensing-based estimates of Evapotranspiration are available at https://doi.org/10.4121/e6e1713a-0c2b-4775-a7f4-9e6e0b2cf40f.v1



**Author contribution**

BT and JvdK conceptualized the review approach; BT, JvdK, SS, MM, and GJ designed the methodology; BT collected and categorized literature; BT and MM conducted data collection for meta-analysis; BT analyzed the data; BT and SS visualized the results; BT wrote the manuscript draft; BT, JvdK, SS, MM, GJ, and RU reviewed and edited the manuscript; GJ, MM and
RU supervised the research activities; MM acquired funding and managed the project.

**Competing interests**

One of the authors is a member of the editorial board of Hydrology and Earth System Sciences.

**Financial support**

This work was supported by the Ministry of Foreign Affairs of the Netherlands through the project "Monitoring land and water
productivity by Remote Sensing (WaPOR phase 2) project" (GCP/INT/729/NET).

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
