# Peer review of "Uncertainty Assessment of Satellite Remote Sensing-based Evapotranspiration Estimates: A Systematic Review of Methods and Gaps"

_EGUsphere, 2023_

## Referee Comment (RC1)

This is a great paper giving an overview of remotely sensed ET evaluation approaches in the literature. It's well-written and interesting. Such an undertaking is certainly a large task so it's understandable that the authors would miss some literature here and there; I've given a few pointers to uncover large missing areas in the literature. That said, I don't know which of the 601 (plus more coming in revision) papers the authors should cite explicitly in the main text versus refer to implicitly within category, but maybe err on the side of adding more in-text references unless EGUsphere pushes back with a limit? Overall, the paper doesn't really have a main result other than that different things are different, but the paper will be a great go-to source for those interested in RS-ET. If scientists follow the recommendations, this could help understand results in a relative context.

Josh Fisher

- There is some discussion on different time scales of analysis, but perhaps some more extensive commentary on instantaneous vs. temporally upscaled validation would be helpful given that most RS-ET is based on polar orbiting instantaneous measurements.
- L31. May want to cite [*Fisher et al.*, 2017].
- L35. May want to cite [*Monteith*, 1965; *Shuttleworth and Wallace*, 1985].
- L39. [*Fisher et al.*, 2017].
- L49. Include ECOSTRESS [*Fisher et al.*, 2020].
- Fig 1. This figure seems to be missing a lot of literature, including reviews cited in the text (e.g., Vinukollu; Jimenez; Melton; etc.).
- L130. "ET is not measured directly by sensors, but is the result from models or reanalyses, and thus…"
- Section 2.3. We used Gaussian Error Propagation in [*Fisher et al.*, 2005] and Method of Moments in [*Fisher et al.*, 2008].
- L185. Period.
- How do you draw the line between diagnostic models, machine learning models, land surface models, etc.? It's sometimes a blurry distinction.
- Figs 5 & 9. I'm not 100% clear on how to read this. It's not obvious what the top bars correspond to. The figure does not label what are the bottom numbers. It's not clear what gray vs. black circles are, and what the connecting lines mean. Maybe define TCH/TH in the caption.
- L243. Curious what are those other approaches?
- Fig 6. Maybe include a secondary y-axis that is the total #.
- Fig 7. I'm not seeing the water balance residual papers here?
- L274. Even smaller with sap flow?
- L308. Slightly misleading because then there was the GRACE-FO mission, which should be mentioned.
- Section 4.1.2. I think you're missing quite a lot of papers here, so you'll have to re-search and update.
- 4.3 out of order.

- Section 4.7. Yunjun Yao and others have been forging forward with many papers in this realm.
- Fig 12. Nice figure.
- L520. Interesting.
- L556. I think it would also depend on the site. If you're using a site with low ET, then your RMSE is likely to be low, and vice versa.
- L581. "in a"
- Section 7. One of the major approaches many of us in the community are working towards is improved spatiotemporal resolution of RS-ET. Moving from ECOSTRESS to SBG, multiple Landsats, TRISHNA, LSTM, and Hydrosat. Would that be worth commenting on here?
- L606. Period.
- L754. Reference repeated.
- Here's a list of more papers to cross-check:

[*McCabe and Wood*, 2006; *Fisher et al.*, 2009; *Glenn et al.*, 2010; *Liang et al.*, 2010; *Blyth and Harding*, 2011; *Fisher et al.*, 2011; *Jiménez et al.*, 2011; *Mueller et al.*, 2011; *Sahoo et al.*, 2011; *Vinukollu et al.*, 2011b; *Vinukollu et al.*, 2011a; *Polhamus et al.*, 2012; *McCabe et al.*, 2013; *Mueller et al.*, 2013; *Polhamus et al.*, 2013; *Armanios and Fisher*, 2014; *Chen et al.*, 2014; *Ershadi et al.*, 2014; *Yao et al.*, 2014; *Chen et al.*, 2015; *Feng et al.*, 2016; *McCabe et al.*, 2016; *Michel et al.*, 2016a; *Michel et al.*, 2016b; *Miralles et al.*, 2016a; *Miralles et al.*, 2016b; *Zhang et al.*, 2016; *Yao et al.*, 2017a; *Yao et al.*, 2017b; *Chang et al.*, 2018; *Jiménez et al.*, 2018; *Xu et al.*, 2018; *Gomis-Cebolla et al.*, 2019; *Guillevic et al.*, 2019; *McCabe et al.*, 2019; *Stoy et al.*, 2019; *Pascolini-Campbell et al.*, 2020; *Sadeghi et al.*, 2020; *Wu et al.*, 2020; *Anderson et al.*, 2021; *Bai et al.*, 2021; *Cawse-Nicholson et al.*, 2021; *Melo et al.*, 2021; *Pascolini-Campbell et al.*, 2021; *Pascolini-Campbell et al.*, 2021; *Shang et al.*, 2021; *Tang et al.*, 2021; *Shi et al.*, 2022; *Xie et al.*, 2022; *Yang et al.*, 2022; *Volk et al.*, 2023]

Anderson, M. C., Y. Yang, J. Xue, K. R. Knipper, Y. Yang, F. Gao, C. R. Hain, W. P. Kustas, K. Cawse-Nicholson, and G. Hulley (2021), Interoperability of ECOSTRESS and Landsat for mapping evapotranspiration time series at sub-field scales, *Remote Sensing of Environment*, *252*, 112189.

Armanios, D. E., and J. B. Fisher (2014), Measuring water availability with limited ground data: assessing the feasibility of an entirely remote-sensing-based hydrologic budget of the Rufiji Basin, Tanzania, using TRMM, GRACE, MODIS, SRB, and AIRS, *Hydrological Processes*, *28*(3), 853-867.

Bai, Y., S. Zhang, N. Bhattarai, K. Mallick, Q. Liu, L. Tang, J. Im, L. Guo, and J. Zhang (2021), On the use of machine learning based ensemble approaches to improve evapotranspiration estimates from croplands across a wide environmental gradient, *Agricultural and Forest Meteorology*, *298*, 108308.

Blyth, E., and R. J. Harding (2011), Methods to separate observed global evapotranspiration into the interception, transpiration and soil surface evaporation components, *Hydrological Processes*, *25*(26), 4063-4068.

Cawse-Nicholson, K., M. C. Anderson, Y. Yang, Y. Yang, S. J. Hook, J. B. Fisher, G. Halverson, G. C. Hulley, C. Hain, and D. D. Baldocchi (2021), Evaluation of a CONUS-wide ECOSTRESS DisALEXI evapotranspiration product, *IEEE Journal of Selected Topics in Applied Earth Observations and Remote Sensing*, *14*, 10117-10133.

Chang, Y., D. Qin, Y. Ding, Q. Zhao, and S. Zhang (2018), A modified MOD16 algorithm to estimate evapotranspiration over alpine meadow on the Tibetan Plateau, China, *Journal of Hydrology*, *561*, 16-30.

Chen, Y., J. Xia, S. Liang, J. Feng, J. B. Fisher, X. Li, X. Li, S. Liu, Z. Ma, and A. Miyata (2014), Comparison of satellite-based evapotranspiration models over terrestrial ecosystems in China, *Remote Sensing of Environment*, *140*, 279-293.

Chen, Y., W. Yuan, J. Xia, J. B. Fisher, W. Dong, X. Zhang, S. Liang, A. Ye, W. Cai, and J. Feng (2015), Using Bayesian model averaging to estimate terrestrial evapotranspiration in China, *Journal of Hydrology*, *528*, 537-549.

Ershadi, A., M. F. McCabe, J. P. Evans, N. W. Chaney, and E. F. Wood (2014), Multi-site evaluation of terrestrial evaporation models using FLUXNET data, *Agricultural and Forest Meteorology*, *187*, 46-61.

Feng, F., X. Li, Y. Yao, S. Liang, J. Chen, X. Zhao, K. Jia, K. Pinter, and J. H. McCaughey (2016), An empirical orthogonal function-based algorithm for estimating terrestrial latent heat flux from eddy covariance, meteorological and satellite observations, *Plos one*, *11*(7).

Fisher, J. B., K. Tu, and D. D. Baldocchi (2008), Global estimates of the land-atmosphere water flux based on monthly AVHRR and ISLSCP-II data, validated at 16 FLUXNET sites, *Remote Sensing of Environment*, *112*(3), 901-919.

Fisher, J. B., R. H. Whittaker, and Y. Malhi (2011), ET Come Home: A critical evaluation of the use of evapotranspiration in geographical ecology, *Global Ecology and Biogeography*, *20*, 1-18.

Fisher, J. B., T. A. Debiase, Y. Qi, M. Xu, and A. H. Goldstein (2005), Evapotranspiration models compared on a Sierra Nevada forest ecosystem, *Environmental Modelling & Software*, *20*(6), 783-796.

Fisher, J. B., F. Melton, E. Middleton, C. Hain, M. Anderson, R. Allen, M. F. McCabe, S. Hook, D. Baldocchi, and P. A. Townsend (2017), The future of evapotranspiration: Global requirements for ecosystem functioning, carbon and climate feedbacks, agricultural management, and water resources, *Water Resources Research*, *53*(4), 2618-2626.

Fisher, J. B., et al. (2009), The land-atmosphere water flux in the tropics, *Global Change Biology*, *15*, 2694-2714.

Fisher, J. B., et al. (2020), ECOSTRESS: NASA's Next Generation Mission to Measure Evapotranspiration From the International Space Station, *Water Resources Research*, *56*(4), e2019WR026058.

Glenn, E., P. Nagler, and A. Huete (2010), Vegetation Index methods for estimating evapotranspiration by remote sensing, *Surveys in Geophysics*, *31*(6), 531-555.

Gomis-Cebolla, J., J. C. Jimenez, J. A. Sobrino, C. Corbari, and M. Mancini (2019), Intercomparison of remote-sensing based evapotranspiration algorithms over amazonian forests, *International Journal of Applied Earth Observation and Geoinformation*, *80*, 280-294.

Guillevic, P. C., A. Olioso, S. J. Hook, J. B. Fisher, J.-P. Lagouarde, and E. F. Vermote (2019), Impact of the revisit of thermal infrared remote sensing observations on evapotranspiration uncertainty—A sensitivity study using AmeriFlux Data, *Remote Sensing*, *11*(5), 573.

Jiménez, C., B. Martens, D. M. Miralles, J. B. Fisher, H. E. Beck, and D. Fernández-Prieto (2018), Exploring the merging of the global land evaporation WACMOS-ET products based on local tower measurements, *Hydrology and Earth System Sciences*, *22*(8), 4513-4533.

Jiménez, C., et al. (2011), Global inter-comparison of 12 land surface heat flux estimates, *Journal of Geophysical Research*, *116*(D02102), doi:10.1029/2010JD014545.

Liang, S., K. Wang, X. Zhang, and Wild, Martin (2010), Review on estimation of land surface radiation and energy budgets from ground measurement, remote sensing and model simulations, *IEEE Journal of Selected Topics in Applied Earth Observations and Remote Sensing*, *3*(3), 225-240.

McCabe, M., et al. (2013), Global-scale estimation of land surface heat fluxes from space: product assessment and intercomparison, in *Remote Sensing of Energy Fluxes and Soil Moisture Content*, edited by G. P. Petropoulos, p. 538, CRC Press, Taylor & Francis Group.

McCabe, M. F., and E. F. Wood (2006), Scale influences on the remote estimation of evapotranspiration using multiple satellite sensors, *Remote Sensing of Environment*, *105*(4), 271-285.

McCabe, M. F., D. G. Miralles, T. R. Holmes, and J. B. Fisher (2019), Advances in the remote sensing of terrestrial evaporation, edited, p. 1138, MDPI.

McCabe, M. F., A. Ershadi, C. Jimenez, D. G. Miralles, D. Michel, and E. F. Wood (2016), The GEWEX LandFlux project: evaluation of model evaporation using tower-based and globally gridded forcing data, *Geoscientific Model Development*, *9*(1), 283-305.

Melo, D., J. Anache, V. Borges, D. Miralles, B. Martens, J. Fisher, R. Nóbrega, A. Moreno, O. Cabral, and T. Rodrigues (2021), Are remote sensing evapotranspiration models reliable across South American ecoregions?, *Water Resources Research*, *57*(11), 1-23.

Michel, D., C. Jiménez, D. Miralles, M. Jung, M. Hirschi, A. Ershadi, B. Martens, M. McCabe, J. Fisher, and Q. Mu (2016a), TheWACMOS-ET project–Part 1: Tower-scale evaluation of four remote-sensing-based evapotranspiration algorithms, *Hydrology and Earth System Sciences*, *20*(2), 803-822.

Michel, D., C. Jiménez, D. G. Miralles, M. Jung, M. Hirschi, A. Ershadi, B. Martens, M. F. McCabe, J. B. Fisher, and Q. Mu (2016b), The WACMOS-ET project–Part 1: Tower-scale evaluation of four remote-sensing-based evapotranspiration algorithms, *Hydrology and Earth System Sciences*, *20*(2), 803-822.

Miralles, D., C. Jiménez, M. Jung, D. Michel, A. Ershadi, M. McCabe, M. Hirschi, B. Martens, A. Dolman, and J. Fisher (2016a), The WACMOS-ET project, part 2: evaluation of global terrestrial evaporation data sets, *Hydrology and Earth System Sciences*, *20*(2), 823-842.

Miralles, D. G., C. Jiménez, M. Jung, D. Michel, A. Ershadi, M. McCabe, M. Hirschi, B. Martens, A. J. Dolman, and J. B. Fisher (2016b), The WACMOS-ET project–Part 2: Evaluation of global terrestrial evaporation data sets, *Hydrology and Earth System Sciences*, *20*(2), 823-842.

Monteith, J. L. (1965), Evaporation and the environment, *Symposium of the Society of Exploratory Biology*, *19*, 205-234.

Mueller, B., M. Hirschi, C. Jimenez, P. Ciais, P. Dirmeyer, A. Dolman, J. Fisher, M. Jung, F. Ludwig, and F. Maignan (2013), Benchmark products for land evapotranspiration: LandFlux-EVAL multi-dataset synthesis, *Hydrology & Earth System Sciences*, *17*, 3707-3720.

Mueller, B., et al. (2011), Evaluation of global observations-based evapotranspiration datasets and IPCC AR4 simulations, *Geophysical Research Letters*, *38*(L06402), doi:10.1029/2010GL046230.

Pascolini-Campbell, M., J. T. Reager, H. A. Chandanpurkar, and M. Rodell (2021), A 10 per cent increase in global land evapotranspiration from 2003 to 2019, *Nature*, *593*(7860), 543-547.

Pascolini-Campbell, M., J. B. Fisher, and J. T. Reager (2021), GRACE-FO and ECOSTRESS synergies constrain fine-scale impacts on the water balance, *Geophysical Research Letters*, *48*(15), e2021GL093984.

Pascolini-Campbell, M. A., J. T. Reager, and J. B. Fisher (2020), GRACE-based mass conservation as a validation target for basin-scale evapotranspiration in the contiguous United States, *Water Resources Research*, *56*(2), e2019WR026594.

Polhamus, A., J. B. Fisher, and K. P. Tu (2012), What controls the error structure in evapotranspiration models?, *Agricultural and Forest Meteorology*, *169*, 12-24.

Polhamus, A., J. B. Fisher, and K. P. Tu (2013), What controls the error structure in evapotranspiration models?, *Agricultural and Forest Meteorology*, *169*, 12-24.

Sadeghi, M., A. Ebtehaj, W. T. Crow, L. Gao, A. J. Purdy, J. B. Fisher, S. B. Jones, E. Babaeian, and M. Tuller (2020), Global estimates of land surface water fluxes from SMOS and SMAP satellite soil moisture data, *Journal of Hydrometeorology*, *21*(2), 241-253.

Sahoo, A. K., M. Pan, T. J. Troy, R. K. Vinukollu, J. Sheffield, and E. F. Wood (2011), Reconciling the global terrestrial water budget using satellite remote sensing, *Remote Sensing of Environment*, *115*(8), 1850-1865.

Shang, K., Y. Yao, S. Liang, Y. Zhang, J. B. Fisher, J. Chen, S. Liu, Z. Xu, Y. Zhang, and K. Jia (2021), DNN-MET: A deep neural networks method to integrate satellite-derived evapotranspiration products, eddy covariance observations and ancillary information, *Agricultural and Forest Meteorology*, *308*, 108582.

Shi, M., J. R. Worden, A. Bailey, D. Noone, C. Risi, R. Fu, S. Worden, R. Herman, V. Payne, and T. Pagano (2022), Amazonian terrestrial water balance inferred from satellite-observed water vapor isotopes, *Nature Communications*, *13*(1), 2686.

Shuttleworth, W. J., and J. S. Wallace (1985), Evaporation from sparse crops—an energy combination theory, *Quarterly Journal of the Royal Meteorological Society*, *111*, 839-855.

Stoy, P. C., T. S. El-Madany, J. B. Fisher, P. Gentine, T. Gerken, S. P. Good, A. Klosterhalfen, S. Liu, D. G. Miralles, and O. Perez-Priego (2019), Reviews and syntheses: Turning the challenges of partitioning ecosystem evaporation and transpiration into opportunities, *Biogeosciences*, *16*(19), 3747-3775.

Tang, L., S. Zhang, J. Zhang, Y. Liu, and Y. Bai (2021), Estimating evapotranspiration based on the satellite-retrieved near-infrared reflectance of vegetation (NIRv) over croplands, *GIScience & Remote Sensing*, 1-25.

Vinukollu, R. K., E. F. Wood, C. R. Ferguson, and J. B. Fisher (2011a), Global estimates of evapotranspiration for climate studies using multi-sensor remote sensing data: Evaluation of three process-based approaches, *Remote Sensing of Environment*, *115*, 801-823.

Vinukollu, R. K., R. Meynadier, J. Sheffield, and E. F. Wood (2011b), Multi-model, multi-sensor estimates of global evapotranspiration: climatology, uncertainties and trends, *Hydrological Processes*, *25*(26), 3993-4010.

Volk, J. M., J. Huntington, F. S. Melton, R. Allen, M. C. Anderson, J. B. Fisher, A. Kilic, G. Senay, G. Halverson, and K. Knipper (2023), Development of a benchmark Eddy flux evapotranspiration dataset for evaluation of satellite-driven evapotranspiration models over the CONUS, *Agricultural and Forest Meteorology*, *331*, 109307.

Wu, G., X. Cai, T. F. Keenan, S. Li, X. Luo, J. B. Fisher, R. Cao, F. Li, A. J. Purdy, and W. Zhao (2020), Evaluating three evapotranspiration estimates from model of different complexity over China using the ILAMB benchmarking system, *Journal of Hydrology*, *590*, 125553.

Xie, Z., Y. Yao, X. Zhang, S. Liang, J. B. Fisher, J. Chen, K. Jia, K. Shang, J. Yang, and R. Yu (2022), The Global LAnd Surface Satellite (GLASS) evapotranspiration product Version 5.0: Algorithm development and preliminary validation, *Journal of Hydrology*, *610*, 127990.

Xu, J., Y. Yao, S. Liang, S. Liu, J. B. Fisher, K. Jia, X. Zhang, Y. Lin, L. Zhang, and X. Chen (2018), Merging the MODIS and landsat terrestrial latent heat flux products using the multiresolution tree method, *IEEE Transactions on Geoscience and Remote Sensing*, *57*(5), 2811-2823.

Yang, J., Y. Yao, C. Shao, Y. Li, J. B. Fisher, J. Cheng, J. Chen, K. Jia, X. Zhang, and K. Shang (2022), A novel TIR-derived three-source energy balance model for estimating daily latent heat flux in mainland China using an all-weather land surface temperature product, *Agricultural and Forest Meteorology*, *323*, 109066.

Yao, Y., S. Liang, X. Li, Y. Hong, J. B. Fisher, N. Zhang, J. Chen, J. Cheng, S. Zhao, and X. Zhang (2014), Bayesian multimodel estimation of global terrestrial latent heat flux from eddy covariance, meteorological, and satellite observations, *Journal of Geophysical Research: Atmospheres*, *119*(8), 4521-4545.

Yao, Y., S. Liang, X. Li, J. Chen, S. Liu, K. Jia, X. Zhang, Z. Xiao, J. B. Fisher, and Q. Mu (2017a), Improving global terrestrial evapotranspiration estimation using support vector machine by integrating three process-based algorithms, *Agricultural and Forest Meteorology*, *242*, 55-74.

Yao, Y., S. Liang, X. Li, Y. Zhang, J. Chen, K. Jia, X. Zhang, J. B. Fisher, X. Wang, and L. Zhang (2017b), Estimation of high-resolution terrestrial evapotranspiration from Landsat data using a simple Taylor skill fusion method, *Journal of Hydrology*, *553*, 508-526.

Zhang, Y., M. Pan, and E. F. Wood (2016), On creating global gridded terrestrial water budget estimates from satellite remote sensing, in *Remote Sensing and Water Resources*, edited, pp. 59-78, Springer.

---

## Author Response (AR1)

**REVIEWER 1**

**Thank you very much for your extensive and constructive suggestions. Our responses are in bold below.**

*This is a great paper giving an overview of remotely sensed ET evaluation approaches in the literature. It's well-written and interesting. Such an undertaking is certainly a large task so it's understandable that the authors would miss some literature here and there; I've given a few pointers to uncover large missing areas in the literature. That said, I don't know which of the 601 (plus more coming in revision) papers the authors should cite explicitly in the main text versus refer to implicitly within category, but maybe err on the side of adding more in-text references unless EGUsphere pushes back with a limit? Overall, the paper doesn't really have a main result other than that different things are different, but the paper will be a great go-to source for those interested in RS-ET. If scientists follow the recommendations, this could help understand results in a relative context.*

**We appreciate your pointers to some interesting articles. Regarding in-text citations, we cited a reference in a sentence where it provides ideas or information that is neither our own nor common knowledge. For some statements, several citations can be used but including all of them could impact the readability of the text. The included articles were used to systematically quantify the categories and not all of them directly provided ideas or information to our text. Therefore, we did not cite all of them in the text.**

*There is some discussion on different time scales of analysis, but perhaps some more extensive commentary on instantaneous vs. temporally upscaled validation would be helpful given that most RS-ET is based on polar orbiting instantaneous measurements.*

**We also find this a very important point. We have added more commentary on upscaled validation in sections 2.2 (L127-133) and 5.2 (L472-477). A more extensive commentary would not fit in the current objective and structure of this paper, so we would refer reader to other papers cited.**

*L31. May want to cite [Fisher et al., 2017].*

**We have reviewed and cited the reference as it provides evidence to this sentence (L31).**

*L35. May want to cite [Monteith, 1965; Shuttleworth and Wallace, 1985].*

**Citations have been added.**

*L39. [Fisher et al., 2017].*

**The citation has been added. The suggested paper is an interesting commentary for readers to refer to.**

*L49. Include ECOSTRESS [Fisher et al., 2020].*

**ECOSTRESS data product has been included.**

*Fig 1. This figure seems to be missing a lot of literature, including reviews cited in the text (e.g., Vinukollu; Jimenez; Melton; etc.).*

**Figure 1 consists of literature review articles only. The purpose is to direct readers to previous literature reviews and distinguish the topics of those literature reviews from our review. We have cross-checked the suggested articles by Vinukollu, Jimenez, and Melton. These are indeed important original research articles that compared different ET products and explored the merging of some products. However, these articles are not literature reviews, we have referred to them in other sections of our review, but not in Figure 1. We did a search for literature review articles on the topics and included a few more reviews in this figure.**

*L130. "ET is not measured directly by sensors, but is the result from models or reanalyses, and thus…"*

**The sentence has been corrected as suggested.**

*Section 2.3. We used Gaussian Error Propagation in [Fisher et al., 2005] and Method of Moments in [Fisher et al., 2008].*

**We have removed mentioning specific methods (e.g., Monte Carlo) in this section since it is meant to be theoretical (L145-L151). For the period that we reviewed (2011-2021), these methods were not used.**

*L185. Period.*

**The sentence has been corrected.**

*How do you draw the line between diagnostic models, machine learning models, land surface models, etc.? It's sometimes a blurry distinction.*

**There have been many literature reviews that categorized diagnostic ET models, which often differ from each other (Courault et al., 2005; Kalma et al., 2008; Wang and Dickinson, 2012; Zhang et al., 2016; Chen and Liu, 2020). The distinction can be blurry when models fit in more than one category. We can distinguish these types:**

**• Diagnostic vs. prognostic: Diagnostic models estimate the values of ET at the time-of-overpass and upscale to longer period. Prognostic models use data assimilation to predict temporally continuous ET (Wang and Dickinson, 2012).**

**• Machine learning models use data-driven algorithms to estimate ET, not explicitly involve physical processes, models are trained with ground data.**

**• Land surface models are models that simulate various processes that occur at the Earth's land surface, which includes ET. ET is not the main output of these models and is constrained by initial states and other modelled variables (not only input data).**

**We consider RS-ET estimates from models that have 2 criteria: (1) aim to estimate ET as the main output (diagnostic) (2) using satellite data as input (satellite remote**

sensing-based). These models fit in the categories reviewed by Courault et al. (2005), Zhang et al. (2016), and Chen and Liu (2020). We have clarified this in section 3.2 (L192 and footnote 2).

*Figs 5 & 9. I'm not 100% clear on how to read this. It's not obvious what the top bars correspond to. The figure does not label what are the bottom numbers. It's not clear what gray vs. black circles are, and what the connecting lines mean. Maybe define TCH/TH in the caption.*

**We have added explanation of these upset plots and TCH/TH in the captions of Figure 5 and 9.**

*L243. Curious what are those other approaches?*

**We recorded those approaches in https://doi.org/10.4121/797dcaff-56e3-45ae-a931-f6f4a3135d26.v2**

**-    Validation of sub-modules in ET models (De la Fuente-Sáiz et al., 2017).**

**-    Comparison of the ET partitioning (not total ET) to evaluate uncertainty due to model parameterization (Miralles et al., 2016).**

**-    Deduction of the analytical relationship between latent heat flux and AOI size in SEBAL to assess uncertainty due to change of spatial support (Tang et al., 2013).**

**-    Using Analysis of Variance (ANOVA) to compare the mean total evaporation estimates for the different land cover types between Landsat 8 and MODIS to assess uncertainty due to input data (Shoko et al., 2015)**

**-    Using temporal patterns of ET per crop type to evaluate compound uncertainty (Sun et al., 2017).**

**-    Using spatial pattern metric and empirical Copula densities to evaluate relative uncertainty (Stisen et al., 2021)**

**Explicitly listing other approaches seems to be beneficial. We mentioned where readers can find this list in Figure 5 Caption. However, we did not discuss them in as much detail as other approaches since they are less used and often in combination with validation or intercomparison.**

*Fig 6. Maybe include a secondary y-axis that is the total #.*

**We have added the time series of the total number of reviewed articles in Figure 6.**

*Fig 7. I'm not seeing the water balance residual papers here?*

**We have added a subplot in Figure 7 to show the water balance residual papers and moved this Figure before 4.1.1. to follow the text.**

*L274. Even smaller with sap flow?*

**Here, we meant in-situ measurement of ET (sum of soil evaporation, transpiration,**

and interception), while sap flow only measures transpiration. We have added L262-263 on the in-situ measurement of ET components.

*L308. Slightly misleading because then there was the GRACE-FO mission, which should be mentioned.*

**The sentence has been rewritten to be more accurate:**

**"However, the TWSA products only cover the period from 2002 with a gap of 11 months from 2017 to 2018 between the GRACE and GRACE-FO missions." (L297-299)**

*Section 4.1.2. I think you're missing quite a lot of papers here, so you'll have to re-search and update.*

**There are quite a significant number of studies that we reviewed that used WB residual as a reference for validation (N=83). We have included the number of papers with the water balance method in the text to signify this (Figure 7). However, we did not cite all papers because the text is about the caveats and potential improvements of the WB method, and not all of them provide insights on these topics.**

**Of course, we do not claim that our list is exhaustive. Missing papers might be due to the title and abstract, the year of publication did not meet the criteria of our systematic literature search. We have also added a comment this issue in L174-175.**

*4.3 out of order.*

**The 'uncertainty propagation' paragraphs has been moved to section 4.5 to be consistent with the order of Figure 5.**

*Section 4.7. Yunjun Yao and others have been forging forward with many papers in this realm.*

**Thank you for pointing out the work by Yao. We have referred to the papers by this author in L434-436. We want to note that this section discusses the use of ensembles to assess uncertainties in RS-ET estimates, not the advancements of methods to generate these ensembles. Therefore, papers that aimed to improve ensemble methods but not use them to evaluate uncertainty in RS-ET estimates were not included. We have also changed the heading of this section to "Using ensemble of RS-ET estimates" to reflect our objective.**

*L556. I think it would also depend on the site. If you're using a site with low ET, then your RMSE is likely to be low, and vice versa.*

**We also thought that RMSE depends on the site. In our meta-analysis, we recorded the average of in-situ ET**

**(https://doi.org/10.4121/e6e1713a-0c2b-4775-a7f4-9e6e0b2cf40f.v1). Unfortunately, too many studies did not report this value, so we don't have sufficient data to compare RMSE with mean ET. Otherwise, it would be an interesting result to test**

this argument. We made a recommendation to report mean ET in validation studies. We have added this explanation to Section 6.2 (L568-571).

*L581. "in a"*

The sentence has been corrected.

*Section 7. One of the major approaches many of us in the community are working towards is improved spatiotemporal resolution of RS-ET. Moving from ECOSTRESS to SBG, multiple Landsats, TRISHNA, LSTM, and Hydrosat. Would that be worth commenting on here?*

Thank you for your suggestion. We have mentioned this development in Section 7 (L599-601).

*L606. Period.*

The sentence has been corrected.

*L754. Reference repeated.*

Duplication has been removed.

*Here's a list of more papers to cross-check:*

*[McCabe and Wood, 2006; Fisher et al., 2009; Glenn et al., 2010; Liang et al., 2010; Blyth and Harding, 2011; Fisher et al., 2011; Jiménez et al., 2011; Mueller et al., 2011; Sahoo et al., 2011; Vinukollu et al., 2011b; Vinukollu et al., 2011a; Polhamus et al., 2012; McCabe et al., 2013; Muelleret al., 2013; Polhamus et al., 2013; Armanios and Fisher, 2014; Chen et al., 2014; Ershadi et al., 2014; Yao et al., 2014; Chen et al., 2015; Feng et al., 2016; McCabe et al., 2016; Michel et al.,2016a; Michel et al., 2016b; Miralles et al., 2016a; Miralles et al., 2016b; Zhang et al., 2016; Yao et al., 2017a; Yao et al., 2017b; Chang et al., 2018; Jiménez et al., 2018; Xu et al., 2018; Gomis-Cebolla et al., 2019; Guillevic et al., 2019; McCabe et al., 2019; Stoy et al., 2019; Pascolini-Campbell et al., 2020; Sadeghi et al., 2020; Wu et al., 2020; Anderson et al., 2021; Bai et al., 2021; Cawse-Nicholson et al., 2021; Melo et al., 2021; Pascolini-Campbell et al., 2021; Pascolini-Campbell et al., 2021; Shang et al., 2021; Tang et al., 2021; Shi et al., 2022; Xie et al., 2022; Yanget al., 2022; Volk et al., 2023]*

Thank you for the extensive list of references. We have cross-checked the list and found that we have already cited 7 articles in the text and 22 of them are included in the categories. Other articles did not meet some of the search criteria (e.g., year of publication, keywords used in title and abstract).

**References**

Alfieri, J.G., Anderson, M.C., Kustas, W.P. and Cammalleri, C., 2017. Effect of the revisit interval and temporal upscaling methods on the accuracy of remotely sensed evapotranspiration estimates. Hydrology and Earth System Sciences, 21(1), pp.83-98. doi:10.5194/hess-21-83-2017

Courault, D., Seguin, B., Olioso, A.: Review on estimation of evapotranspiration from remote sensing data: From empirical to numerical modeling approaches. Irrig Drainage Syst 19, 223–249. https://doi.org/10.1007/s10795-005-5186-0, 2005.

De la Fuente-Sáiz, D., Ortega-Farías, S., Fonseca, D., Ortega-Salazar, S., Kilic, A., & Allen, R. (2017). Calibration of METRIC Model to Estimate Energy Balance over a Drip-Irrigated Apple Orchard. Remote Sensing, 9(7), 670. doi:10.3390/rs9070670

Gentine, P., Entekhabi, D., Chehbouni, A., Boulet, G. and Duchemin, B., 2007. Analysis of evaporative fraction diurnal behaviour. Agricultural and forest meteorology, 143(1-2), pp.13-29. https://doi.org/10.1016/j.agrformet.2006.11.002

Hoedjes, J.C.B., Chehbouni, A., Jacob, F., Ezzahar, J. and Boulet, G., 2008. Deriving daily evapotranspiration from remotely sensed instantaneous evaporative fraction over olive orchard in semi-arid Morocco. Journal of Hydrology, 354(1-4), pp.53-64. https://doi.org/10.1016/j.jhydrol.2008.02.016

Jiang, L., Zhang, B., Han, S., Chen, H. and Wei, Z., 2021. Upscaling evapotranspiration from the instantaneous to the daily time scale: Assessing six methods including an optimized coefficient based on worldwide eddy covariance flux network. Journal of Hydrology, 596, p.126135. https://doi.org/10.1016/j.jhydrol.2021.126135

Kalma, J.D., McVicar, T.R., McCabe, M.F.: Estimating Land Surface Evaporation: A Review of Methods Using Remotely Sensed Surface Temperature Data. Surv. Geophys. 29, 421–469. https://doi.org/10.1007/s10712-008-9037-z, 2008.

Lex A., Gehlenborg N., Strobelt H., Vuillemot R., Pfister H.: UpSet: Visualization of Intersecting Sets IEEE Transactions on Visualization and Computer Graphics (InfoVis), 20(12): 1983--1992, https://doi.org/10.1109/TVCG.2014.2346248, 2014

Liu, Z., 2021. The accuracy of temporal upscaling of instantaneous evapotranspiration to daily values with seven upscaling methods. Hydrology and Earth System Sciences, 25(8), pp.4417-4433. https://doi.org/10.5194/hess-25-4417-2021

Miralles, D. G., Jiménez, C., Jung, M., Michel, D., Ershadi, A., McCabe, M. F., … Fernández-Prieto, D. (2016). The WACMOS-ET project – Part 2: Evaluation of global terrestrial evaporation data sets. Hydrology and Earth System Sciences, 20(2), 823–842. doi:10.5194/hess-20-823-2016

Shoko, C., Clark, D., Mengistu, M., Dube, T., & Bulcock, H. (2015). Effect of spatial resolution on remote sensing estimation of total evaporation in the uMngeni catchment, South Africa. Journal of Applied Remote Sensing, 9(1), 095997. doi:10.1117/1.jrs.9.095997

Stisen, S., Soltani, M., Mendiguren, G., Langkilde, H., Garcia, M., & Koch, J. (2021). Spatial Patterns in Actual Evapotranspiration Climatologies for Europe. Remote Sensing, 13(12), 2410. doi:10.3390/rs13122410

Sun, L., Anderson, M. C., Gao, F., Hain, C., Alfieri, J. G., Sharifi, A., … McKee, L. (2017). Investigating water use over the Choptank River Watershed using a multisatellite data fusion approach. Water Resources Research, 53(7), 5298–5319. doi:10.1002/2017wr020700

Tang, R., Li, Z.L., Chen, K.S., Jia, Y., Li, C. and Sun, X., 2013. Spatial-scale effect on the SEBAL model for evapotranspiration estimation using remote sensing data. Agricultural and forest meteorology, 174, pp.28-42.

Van Niel, T.G., McVicar, T.R., Roderick, M.L., van Dijk, A.I., Beringer, J., Hutley, L.B. and Van Gorsel, E., 2012. Upscaling latent heat flux for thermal remote sensing studies: Comparison of alternative approaches and correction of bias. Journal of Hydrology, 468, pp.35-46. https://doi.org/10.1016/j.jhydrol.2012.08.005

Wang, K., Dickinson, R.E.: A review of global terrestrial evapotranspiration: Observation, modeling, climatology, and climatic variability. Rev. Geophys. 50. https://doi.org/10.1029/2011RG000373, 2012.

Xu, T., Liu, S., Xu, L., Chen, Y., Jia, Z., Xu, Z. and Nielson, J., 2015. Temporal upscaling and reconstruction of thermal remotely sensed instantaneous evapotranspiration. Remote Sensing, 7(3), pp.3400-3425. https://doi.org/10.3390/rs70303400

Zhang, K., Kimball, J.S., Running, S.W.: A review of remote sensing based actual evapotranspiration estimation. Wiley Interdisciplinary Reviews: Water 3, 834–853. https://doi.org/10.1002/wat2.1168, 2016.

Zhang, X., Wu, J., Wu, H., Chen, H. and Zhang, T., 2013. Improving temporal extrapolation for daily evapotranspiration using radiation measurements. Journal of Applied Remote Sensing, 7(1), pp.073538-073538. https://doi.org/10.1117/1.JRS.7.073538

**REVIEWER 2**

**Thank you very much for your extensive and constructive suggestions. Our responses are in bold below.**

**\*Major comments\***

*1. Due to the nature of a systematic review, it is difficult to differentiate between articles that evaluate the performance of existing ET products and ET-based models. It would be very beneficial to clarify the distinction between evaporation products and the models used to estimate ET. Currently, it is challenging for readers to differentiate between them, making it difficult to follow certain ideas. For instance, in Line 231, the authors discuss eight topics for assessing uncertainty in RS-ET, where some points relate to the evaluation of ET products while others to the models. It would be beneficial to clearly indicate what is defined as RS-ET inthe manuscript and which results are from models or open-acces gridded products.*

**We have clarified the definition of RS-ET in section 3.2 (L191-192). The distinction between products and models is also made in Section 1 (L41-44 and L49-52). The 8 topics discussed in Section 4 are approaches to assess uncertainty in RS-ET estimates, either from model simulations or analysis-ready data products.**

*2. The article is lengthy, and it would be beneficial to condense the sections "Theoretical frameworks" and "Systematic quantitative literature review method" for brevity.*

**We have revised and reduced the text in these sections.**

*3. The manuscript could benefit from discussing which methods and products perform better in specific contexts. It would be helpful to provide insights on the performance of models and products in relation to specific regions, climates, and relevant factors. For example, i) identifying the errors associated with each method/product; ii) the reported advantages and disadvantages of different models/products; iii) important parameters that drive the estimation of ET in existing models; iv) lessons learned from previous evaluations; and v) which models/products have demonstrated higher physical consistency.*

**The suggested topics are important. However, they were not the objectives of this manuscript. Our goal in this study is to investigate the status of the various methods applied for uncertainty assessment of RS-ET estimates, discuss the advances and caveats of these methods, identify assessment gaps, and provide recommendations for future assessment. Our argument is that because these models and products are evaluated using different assessment methods and reference data, it is not reliable to rank their performance and generalize the conclusion to all contexts.**

**Furthermore, many literature reviews (Figure 1) have discussed some of these topics repetitively:**

**i)   identifying the errors associated with each method/product**

**ii)   the reported advantages and disadvantages of different models/products**

**iii)   important parameters that drive the estimation of ET in existing models**

**Regarding the performance of models and products in relation to specific regions,**

climates, and relevant factors and v) which models/products have demonstrated higher physical consistency, we prefer not to draw conclusions from the reviewed literature because not all models have been compared simultaneously. We do think that this could be the focus of a different paper.

However, we do think that "iv) lessons learned from previous evaluations" could be relevant to our manuscript. We have discussed some in section 7. We have also emphasized the role of developing uncertainty assessment methods to investigate the other topics that you mentioned (L612-613).

*4. In the section "Review of methods for RS-ET uncertainty assessment", the authors could focus on the performance of models/products and relate their findings to specific regions and climates when reported. Addressing questions such as which models performed better in certain areas and why, the sources of uncertainty, the relevance of spatio-temporal resolution in operational applications, the impact of geographical features on model/product uncertainty, and the influence of climate on product performance would greatly enhance this section.*

Section 4 "review methods for RS-ET uncertainty assessment" focuses on the methods of uncertainty assessment (how each method was applied in reviewed literature), not the results of those assessments per se, which is more discussed in Section 6. Therefore, we don't think focusing on the performance of models/products and their relation to specific regions and climates should be the focus of this section. As we point out in our response above, this is clearly an important topic that could be addressed by another article or even a special issue.

We want to emphasize that other literature reviews (Figure 1) focused on the performance of RS-ET models/products, while our review discusses the methods to assess them, as we have outlined in the research questions. The research questions suggested by the reviewers are important to investigate. However, our methods did not aim to answer these questions (i.e., Which models performed better in certain areas and why, the impact of geographical features on model/product uncertainty, and the influence of climate on product performance) and consequently our results do not address them. There are gaps in uncertainty assessment in terms of geographical regions and models, and also inconsistency in methods (Sections 5 and 6). We believe that it is unfair to conclude which models performed better based on the current literature (L607-610). Furthermore, in agreement with reviewer 1, the RMSE depends on the value of ET (568-569).

We have discussed the sources of uncertainty in section 2.2 and their uncertainty assessment (section 5.2).

The relevance of spatio-temporal resolution in operational applications is mentioned in L128-132.

*5. Consider reducing the use of acronyms that are infrequently mentioned in the manuscript, as it can improve readability and comprehension*

We agree that some acronyms are not frequently and necessarily used. We have reduced the use of these acronyms

- **Essential Climate Variables (ECVs)**

- **Monte Carlo method (MCM), not in Section 2.**

- **Sensitivity Analysis (SA)**

- **Systematic Quantitative Literature Review (SQLR)**

- **Web of Science (WoS)**

*6. The authors should clarify the timeframe of their study. While they mention focusing on the period from 2011, the end date or year is only specified on L187 stating that the databases were last accessed on 21.09.2021. It would be very valuable to update the research up to a more recent date to provide a comprehensive evaluation :)*

**We have added articles from 21.09.2021 until the end of 2021 to have a complete year. The last access date has been changed to 24/07/2023. As the body of literature is huge and growing faster than ever (Annex 2), there will always be a gap between the last date of articles accessed and the most recent literature by the time the analysis is complete. In addition, after adding 75 articles more in the literature categories, we did not notice any significant changes in our results. Therefore, we consider more than 600 articles and one decade of recent literature (until 2021) is extensive and comprehensive enough to provide conclusions to our research objectives.**

*7. The authors mention using keywords like "accuracy," "bias," and "precision" to assess uncertainty in products, although these terms differ from the proper definition of uncertainty. It would be important to include the term "performance" in the evaluation, as many studies summarize their findings in terms of model or product performance.*

**We are not sure if this comment relates to the search terms in Table 1. The definitions of these terms are indeed different from 'uncertainty' but, as discussed in Section 2.1, they are used by various authors to describe uncertainty. The variants of 'uncertainty' keyword were selected by iterating our search several times until the results include all the articles in Supplementary Information Annex 1. Since we combined these terms with "OR" in our search query, we have included all the articles that use either one or more of these terms.**

**We acknowledge the term "performance" is also often used. Therefore, we have done an additional search to include "performance" in keywords and found 34 articles more (+7.6%) for 2011-2020.**

*8. Section 6, "Results of RS-ET uncertainty assessment," primarily evaluates articles based on RMSE. However, comparing articles solely on RMSE is not very meaningful, as this goodnes-of-fit metric does not allow for comparisons across areas with different climates and ET patterns. Therefore, the metrics presented in Table 4 (median, mean, quartiles, standard deviations) and Figure 14, which are grouped by evaluated temporal scale may be misleading. A good and valuable reccomentation that the athors could use in their artile could be related to the fact that researchers should report uncertainty/performance metrics using indices that are comparable across studies and not influenced by regional climate or specific ET patterns. It would be valuable to discuss about which metric is reported to be better proxy of model/product performance and which models/products performed better.*

**We did not aim to compare articles or models/products based on RMSE. We agree**

that it is not fair to compare models/products across areas with different climates and ET patterns using solely RMSE. The purpose of Figure 14 and Table 4 is to identify the typical range of reported uncertainty in RS-ET estimates globally (our third research question). The spreading of the RMSE is partially due to the effect of different climates and site-specific conditions. This is why we did not use our results to conclude on which models/products perform better or worse than any others. We have clarified this important issue in Section 6.2 (L568-571, L583-584, and 589-591).

We find that grouping the reported RMSE by temporal scale is valuable to show the effect of temporal upscaling across hundreds of studies (L558-560).

It is indeed valuable to report uncertainty using metrics that are comparable across studies, in order to assess which models/products perform better in different context. We have added this to our recommendations in Section 7 (L626-627).

*Minor comments*

*L10: The authors can emphasise here that evapotranspiration is often referred to as evaporation. As it is currently written, it seems that the authors are referring to both evaporation and evapotranspiration.*

The sentence has been rewritten as "Satellite remote sensing (RS) data are increasingly being used to estimate total evaporation, often referred to as evapotranspiration (ET), over large regions."

*L39-42: Here, the authors mention some methods, but the list is not exhaustive. They can add GLEAM to this list for example, which is a well-known method that drives a ET product with the same name.*

The list is definitely not exhaustive. We have added GLEAM and PT-JPL, which are very frequently used in the included literature.

*L44-45: This sentence can be rewritten for better clarity.*

*L46: The authors mention that retrieving ET estimates from some models requires expertise about the models. However, this is true for every model, so this sentence can be deleted.*

L44-46 has been be rewritten for clarity: "Furthermore, retrieving ET estimates requires access to the data, software or source code, and expertise in these models. The limited accessibility of RS-ET models leads to significant challenges to operational applications of RS-ET estimates (e.g., irrigation scheduling and drought monitoring)."

*L50: This sentence is a bit convoluted. It would be helpful if the authors could clarify their intended meaning.*

L50 has been be rewritten as "Given that more RS-ET data products are becoming available, information about the uncertainties in RS-ET estimates is important for data users (i.e., water managers and policymakers) to apply them properly." (L53)

*L51-53: Here, the authors mention that uncertainty assessment helps data users determine the level of confidence they can have in ET estimates and inferred information about water resources. Since readers of this article may be researchers exploring ET products and models for the first time, it would be a good idea to mention that the use of the products is also limited by their spatio-temporal resolution, specific applications, and latency.*

**Good point. We have added this sentence "Inferences based on RS-ET data products are also limited by their spatio-temporal resolution, latency, and specifications." (L56-57)**

*L55: "foci" should be changed to "focus." The focus of multiple articles is explained in Table S2.*

**Since we mean to say that each of these reviews has a different focus, we want to keep the plural form of the word. But we understand that this collocation of words might sound odd to some readers. We have changed "foci" to "main topics".**

*L59: What do the authors mean by "spatial data production"?*

**We mean the generation of spatial data, which also covers methods other than remote sensing.**

*L60: What do the authors mean by "a good practice protocol for operational validation"?*

**An operational validation workflow as defined by Bayat et al. (2021) has four components, one of which is based on a good practice protocol for validation agreed upon by the community. A good practice protocol for validation is a set of guidelines that are known to produce reliable validation results. For example, the authors have pointed to good practice protocol for validation of Land Surface Temperature (Guillevic et al., 2018), Surface Albedo (Wang et al., 2019), Leaf Area Index (Fernandes et al., 2014), Soil Moisture (Gruber et al., 2020).**

*L59: What do the authors mean by "complete documentation"?*

**Documentation of the ET estimation that provides sufficient information for data users to judge the accuracy and representativeness of the estimates. Allen et al. (2011) have recommended which information to be included in such documentation.**

*Figure 1: This figure is very good and helpful. It will surely assist readers in accessing previous literature review articles. Could the authors complete the list of existing manuscripts related to the review of RS-ET estimation, uncertainty, and validation of products (and models)?*

**Figure 1 consists of only literature review articles. The purpose is to direct readers to previous literature reviews and distinguish the topics of those literature reviews from our review. We have extended the list with more relevant review articles.**

*L130: "reanalyzes" should be "reanalyses."*

**We have changed to "reanalyses".**

*Additionally, could the authors rewrite this sentence to better explain what is considered a high level of processing?*

**By 'level of processing', we meant that they are model output or results from analyses of less processed data and we referred to data user guides by ESA and NASA. The sentence has been rewritten as followed:**

**"ET is not directly measured by sensors but derived from models, thus, considered high-level processing by data providers (ESA, 2021; NASA, 2021). The retrieval models of low-level data (e.g., radiance, vegetation indices) share common formulas and usually requires only raw satellite images. High-level data like RS-ET relies on various models with different concepts, assumptions and data sources."**

*L143: What about replacing "true" with "more accurately representing the ET values"?*

**This sentence has been rewritten:**

**"Modeled estimates are typically validated against a more accurate reference."**

*Figure 3: Please replace "support" with "resolution."*

**We understand why "resolution" is suggested because it is linked to the resampling of RS data. 'Resolution' is how detailed RS data is, measured by the size of the pixel. While 'support' is the volume, shape, size, and orientation that measurement represents. In RS data, these two are similar because the support of ET value in a pixel is also the size of that pixel. However, we wanted to use "support" here because when ET estimates are derived from RS data or validated with reference data, uncertainty occurs also due to a 'change of spatial support' from the pixel size to the footprint size of the measurement. We have changed to "scale" because this term is more general and includes both "resolution" and "support" (also "extent" and "spacing") (Bloschl and Sivapalan, 1995). We have also added footnote 1 to clarify the terminologies in the text of Section 2.2 (L133).**

*Why does the model calculation not have a number? There is uncertainty regarding whether the model is able to resemble physical processes or not.*

**In remote sensing literature, "uncertainty regarding whether the model is able to resemble physical processes or not" is less often acknowledged (Povey et Grainger, 2015; Foody and Atkinson, 2003) unlike in hydrological modeling (Liu and Gupta, 2007; Nearing et al., 2014). This is due to the fact that RS retrieval models usually share common concepts or formulas, especially for low-level data products (e.g., Surface Radiance, NDVI). Since we have argued before that high-level RS data such as ET are outputs of models that often have different concepts**

and assumptions (e.g., SEB vs. PM), we should indeed include uncertainty from the 'model conceptualization', especially for RS-ET processing chain. We have added "model conceptualization" linked with "model calculation" in the figure. We have also added this explanation to Section 2.2 (L123-126).

*Finally, the authors can mention in the figure that compound uncertainty is the sum of all other uncertainties.*

**We have added the explanation of compound uncertainty in Figure 3 caption.**

*L153: Why specifically refer to Monte Carlo when there are more advanced techniques to assess uncertainty propagation?*

**It is the method we observed most frequently when reviewing the literature. As we revised Section 2, we find listing specific methods is not necessary as this section is meant to be theoretical. Therefore, we have removed this mention of Monte Carlo, and only mentioned in Section 4.5 (result of literature review).**

*L165: Can the authors add a sentence on how the definition of validation has changed over time?*

**The sentence has been rewritten as "The definition of validation in modeling is context-dependent and has become more well-defined over time (Bellocchi et al., 2011)."**

*L170: This sentence is not very clear to me. What do the authors mean by "model validation and data" in this context? Maybe the parentheses are misplaced and disrupt the flow of the sentence?*

*L171-172: Can this sentence be deleted? I think the idea is clearly explained in the following sentences.*

**These sentences have been rewritten to improve clarity and briefness (L160-165).**

*L176: This sentence could be rewritten for clarity. Something like: "Validating a model used to derive ET estimates does not necessarily imply that it can be used with different forcing data and provide accurate results. Therefore, when a model is applied to derive ET estimates with different forcings or in different settings, its performance must be evaluated." In the current version, it is difficult to disentangle what is a model, an ET product, and a product based on running the model with different forcings :)*

**Thank you for your suggestion. We have rewritten the sentence as "Because RS-derived data products are model results, their validation depends on the quality and quantity of input parameters and the accuracy of auxiliary hypotheses that were used to derive them (Oreskes et al., 1994). Therefore, validating a RS-ET model does not imply that the model can be applied with any forcing data or**

settings to produce accurate output." (L165-168). Also, in the introduction, we have clarified what we mean by "data product" (L49-52).

*L183: "by" instead of "tby"*

**This has been corrected.**

*L189-190 and Table 1: It would be interesting to know how these terms were chosen. What about other terms like "performance," "quality" (alone), and "error"?*

**We explained this in L179-181. As mentioned before, we have included the term "performance" in a new search.**

*L200: "process" instead of "system."*

**We have changed that.**

*Figure 4: What does "not using the same method to report uncertainty" mean?*

**For metanalysis, we wanted to include studies that assess uncertainty using the same approach (validation), reference data (Eddy Covariance), and metrics. We have added this explanation to the caption.**

*Figure 5 and 9: Why are there 38 articles without any link to a topic? Could the authors provide an explanation in the caption?*

**Thank you very much for pointing this out. We realized that these are the articles excluded after scanning full-text, which is why they are not linked with any topic. We made the mistake of not excluding them when visualizing the dataset. We have corrected this in both figures.**

*Figure 6: It is difficult to see the low values on the graph. Maybe consider using a barplot to visualise this more straightforwardly.*

**We have changed to bar plot to make the low values more visible.**

*L256: "Estimate ET" instead of "observe ET." Remember that ET cannot be directly observed ;)*

**Indeed. We have changed that.**

*Figure 7: I really liked this figure! In the caption, the authors can add an explanation of the "others" category. Are irrigation and water balance articles combined in this category?*

**Thank you. We have added an explanation of the "others" category. The irrigation water balance is different from the catchment water balance (Section 4.1.2). These papers used measurements about rainfall, irrigation, and drainage of agricultural plots to derive ET and did not use a lysimeter, so we put them in a different category. We have added a subplot in this figure to show that "Field Water Balance" category is a part of "In-situ ET estimation" and different from "ET derived from catchment water balance".**

*L303: Here, the authors could briefly mention the assumptions of the simplified water balance.*

**We have added that to the text (L291-295).**

*L309: Still less known compared to what? Maybe rephrase the sentence to clarify.*

**This sentence has been rewritten as "Some techniques have been developed to reconstruct this gap in the GRACE time series (e.g., Yang et al., 2021). However, the uncertainties in gap-filled dS/dt estimates is still less known than uncertainties in the initial estimates from GRACE and GRACE-FO (Boergens et al., 2022)." (L299-301)**

*L342: Some acronyms are introduced more than once, e.g., SA.*

**We have removed acronyms that were introduced more than once.**

*L447: Maybe consider renaming this subsection to something other than "Research Objectives." For example: "Assessment based on the objectives of the analysed manuscripts."*

**Indeed, the subsection heading does sound a little confusing. We have changed it to "Objectives of the reviewed articles"**

*L580: There is a missing space between "in" and "a."*

**The sentence has been corrected.**

*L580-581: I completely agree that further research should combine local and global evaluation efforts, but including a reason for this in the text could be very beneficial for the readers.*

**We have added our explanation in L612-616.**

*L593-593: I do not completely agree with this argument. The RMSE ranges could serve as a baseline, but we have to keep in mind that they are not directly comparable.*

**We have rewritten this sentence as "The RMSE range reported in our study should be only used as a baseline for future studies that validate RS-ET estimates using EC.", and also emphasized that "While RMSE stands as the most commonly employed metric in the literature, it is unsuitable for comparing uncertainties in RS-ET across different studies due to its inherent scale-dependency." (L607-610)**

*L601: What do the authors mean by "matched as much as possible"?*

**We have rewritten this sentence to improve clarity as follows "RS-ET estimates should be converted to values at the temporal and spatial scale of reference datasets."**

*L602: I do not completely agree with this statement. We should report metrics that enable a fair comparison between regions with different climates/patterns.*

**We agree that to compare uncertainties of ET between regions with different climates (thus, different ranges of ET), we need to use scale-independent metrics. We will rewrite the recommendations as follows:**

**•	The four common metrics (RMSE, bias/mean error, correlation coefficient, coefficient of determination) and mean ET should be reported in validation studies.**

**•	In addition, uncertainties in RS-ET estimates should be characterized using multiple metrics that are scale-independent to facilitate comparison of RS-ET uncertainty across regions with different ET ranges.**

*L605: What do the authors mean by this statement? Please provide further clarification.*

**We have rewritten the statement to improve clarity: "Validation of RS-ET models and data products should be reported at different levels of spatial and temporal scales, covering multiple locations."**

*L611-613: I did not understand this sentence. Could the authors please provide additional clarification or rephrase the sentence for clarity?*

**We have rewritten this part as follows: "Several studies have aimed to offer spatially explicit uncertainty in thematic classification, such as land cover and soil type. These studies, like the ones mentioned by Woodcock (2002), have primarily focused on qualitative mapping techniques. However, for quantitative remote sensing, which involves mapping continuous variables like ET, there is a need for methods that can effectively characterize spatially explicit uncertainty. Therefore, we strongly recommend the development and application of methods to evaluate spatiotemporal uncertainty in RS-ET datasets."**

**References**

Allen, R.G., Pereira, L.S., Howell, T.A. and Jensen, M.E., 2011. Evapotranspiration information reporting: II. Recommended documentation. Agricultural Water Management, 98(6), pp.921-929.

Blöschl, G. and Sivapalan, M., 1995. Scale issues in hydrological modelling: a review. Hydrological processes, 9(3-4), pp.251-290. https://doi.org/10.1002/hyp.3360090305

Boergens, E., Kvas, A., Eicker, A., Dobslaw, H., Schawohl, L., Dahle, C., Murböck, M. and Flechtner, F., 2022. Uncertainties of GRACE-Based Terrestrial Water Storage Anomalies for Arbitrary Averaging Regions. Journal of Geophysical Research: Solid Earth, 127(2), p.e2021JB022081.

Gruber, A., De Lannoy, G., Albergel, C., Al-Yaari, A., Brocca, L., Calvet, J.C., Colliander, A., Cosh, M., Crow, W., Dorigo, W. and Draper, C., 2020. Validation practices for satellite soil moisture retrievals: What are (the) errors?. Remote sensing of environment, 244, p.111806.

Liu, Y.Q. and Gupta, H.V., 2007. Uncertainty in hydrologic modeling: toward an integrated data assimilation framework. Water Resources Research, 43 (7), W07401. doi:10.1029/2006WR005756

P. Guillevic, F. Göttsche, J. Nickeson, M. Román (Eds.), Best Practice for Satellite-Derived Land Product Validation, Land Product Validation Subgroup (WGCV/CEOS (2018), p. 58, doi: 10.5067/doc/ceoswgcv/lpv/lst.001

R.A. Fernandes, S.E. Plummer, J. Nightingale, F. Baret, F. Camacho, H. Fang, S. Garrigues, N. Gobron, M. Lang, R. Lacaze, S.G. Leblanc, M. Meroni, B. Martinez, T. Nilson, B. Pinty, J. Pisek, O. Sonnentag, A. Verger, J.M. Welles, M. Weiss, J.-L. Widlowski, G. Schaepman-Strub, M.O. Román, J. Nicheson. Global Leaf Area Index Product Validation Good Practices. CEOS Working Group on Calibration and Validation - Land Product Validation Sub-Group (2014), doi:10.5067/doc/ceoswgcv/lpv/lai.002

Z. Wang, J. Nickeson, M. Román (Eds.), Best Practice for Satellite Derived Land Product Validation, Land Product Validation Subgroup (WGCV/CEOS) (2019), p. 45, doi: 10.5067/DOC/CEOSWGCV/LPV/ALBEDO.001

**REVIEWER 3**

**Thank you very much for your comments and suggestions. Our responses are in bold below.**

*The manuscript surveyed and reviewed the status of the various methods used for uncertainty assessment of remote sensing based estimation of evapotranspiration. It discussed the advances and caveats of the different methods, identified assessment gaps, and provided recommendations for future studies.*

*This reviewer considers such an assessment very useful for the community in using the various RS-ET estimates in hydrological studies. It feels however that some important aspects are missing which concern the model physics and dynamics and the considered physical processes in estimating ET using remote sensing data as input. The urgent challenge to the hydrological remote sensing community is therefore investigating the physics and dynamics of the processes involved in evapotranspiration and devising adequate methods to represent such processes in generating the RS-ET estimates. Once a chosen model is able to adequately represent such physics and dynamics for a few quality controlled reference in-situ sites, the uncertainty in their application to other sites and the globe is considerably reduced, because we can confidently expect that the physics is the same everywhere and the dynamics can be attributed to the temporal resolution of the model and the input data.*

**Indeed, it is very important to investigate the physics and dynamics of the processes involved in ET. However, that is not the intention of this paper. Our premise is that given the availability of satellite data, we have the opportunity to estimate ET based on its relationship with variables that are observable from satellites. There have been many models developed to represent processes (physically-based) or to derive ET from data (empirical or semi-empirical), as reviewed by many authors (Figure 1 in this paper). However, the methods to evaluate the uncertainty of these models are not consistent (this paper).**

**Regardless of the model physics, assessment of uncertainty in RS-ET estimates is needed for the end-users of these estimates. Here, we are considering the uncertainty in RS-ET estimates, which depends not only on the model physics but also the input data. As mentioned in L170-175, if a model is validated in a few sites, the uncertainty in RS-ET outputs in other sites with different characteristics can be different.**

**It is challenging to assess uncertainty everywhere with only a few in-situ sites. The physics is expected to be the same everywhere, but the dominant processes and factors are not the same everywhere (Zhang et al., 2016). The quality of RS observations is not the same everywhere due to spatially varied atmospheric conditions. The quality of meteorological input data is also not the same everywhere. Therefore, we recommend that multiple assessment methods are needed. This will help understand better whether the uncertainty can be attributed to input or model.**

*Fig. 14 needs some more explanation for the different symbols (this is obviously a box plot, but it is not clear to the reader by itself what the different statistics are compared to Table 4).*

**Thank you for pointing this out. We have added a legend for the boxplot and probability density curve in Figure 14 that explains their relations with Table 4.**